# Chromosomal inversion polymorphisms shape the genomic landscape of deer mice

**Olivia S. Harringmeyer** ✉ **and Hopi E. Hoekstra** ✉

Chromosomal inversions are an important form of structural variation that can affect recombination, chromosome structure and fitness. However, because inversions can be challenging to detect, the prevalence and hence the significance of inversions segregating within species remains largely unknown, especially in natural populations of mammals. Here, by combining population-genomic and long-read sequencing analyses in a single, widespread species of deer mouse (*Peromyscus maniculatus*), we identified 21 polymorphic inversions that are large (1.5–43.8 Mb) and cause near-complete suppression of recombination when heterozygous (0–0.03 cM Mb$^{-1}$). We found that inversion breakpoints frequently occur in centromeric and telomeric regions and are often flanked by long inverted repeats (0.5–50 kb), suggesting that they probably arose via ectopic recombination. By genotyping inversions in populations across the species' range, we found that the inversions are often widespread and do not harbour deleterious mutational loads, and many are likely to be maintained as polymorphisms by divergent selection. Comparisons of forest and prairie ecotypes of deer mice revealed 13 inversions that contribute to differentiation between populations, of which five exhibit significant associations with traits implicated in local adaptation. Taken together, these results show that inversion polymorphisms have a significant impact on recombination, genome structure and genetic diversity in deer mice and likely facilitate local adaptation across the widespread range of this species.

A longstanding goal in population genetics has been to quantify intraspecific genetic variation, which serves as the substrate for evolutionary change. Since Lewontin and Hubby first characterized protein sequence variation in *Drosophila pseudoobscura* in 1966, tremendous progress has been made in measuring levels of single nucleotide polymorphisms (SNPs) in a wide diversity of species[1]. However, the prevalence of structural genomic variation, a focus of cytogenetics, remains largely uncharacterized in the molecular era[2]. Chromosomal inversions, in particular, are an important form of structural variation: inversions can be large (affecting megabases of sequence)[3] and have been implicated in local adaptation, including differentiation of annual

and perennial ecotypes of monkeyflowers[4], wing-pattern morphs of mimetic butterflies[5] and mating types of ruffs[6,7].

Inversions may play a key role in local adaptation because of their effects on recombination. When heterozygous, an inversion will suppress recombination with the noninverted arrangement and, as a result, can drastically increase linkage disequilibrium (LD) between the loci it carries. As such, inversions can act as 'supergenes'[8], linking multiple locally adaptive alleles together into coinherited haplotype blocks, which may be advantageous in the face of gene flow[9–11]. Although inversions have been identified across a diversity of species in the context of local adaptation, suggesting that beneficial inversions may

Department of Organismic & Evolutionary Biology, Department of Molecular & Cellular Biology, Museum of Comparative Zoology and Howard Hughes Medical Institute, Harvard University, Cambridge, MA, USA. ✉e-mail: olivia_meyerson@g.harvard.edu; hoekstra@oeb.harvard.edu

be common[3], few studies have performed unbiased scans across the genome for inversion polymorphisms (but see refs. [12–16]), raising the question of whether adaptive inversions are the exception or the rule. Thus, characterizing the abundance of inversion polymorphisms—that is, inversions segregating within a species—is a critical step towards quantifying levels of intraspecific genetic variation and understanding how and why inversion polymorphisms are established and maintained.

Detecting inversion polymorphisms with molecular data has traditionally been challenging (for example, breakpoints often reside in highly repetitive regions)[17], but recent advances in long-read sequencing and increased feasibility of population-level genome resequencing provide new, powerful approaches for identifying inversions[18,19]. Using these approaches, recent studies have revealed the abundance of inversion polymorphisms in a few species: for example, sunflowers harbour dozens of large (1–100 Mb) inversion polymorphisms[16], and humans harbour, on average, hundreds of inversion polymorphisms that affect more DNA base pairs (bp) in total than SNPs[12,13].

Here, we perform an unbiased genome-wide scan for inversion polymorphisms in the deer mouse, *Peromyscus maniculatus*. The deer mouse is the most abundant and widespread mammal in North America: it has large effective population sizes[20,21] and a range spanning all major terrestrial habitats, including dense forests and open prairies[22]. Early cytogenetic work in deer mice identified at least 13 visible chromosomal rearrangements[23,24]. Returning to this system in the molecular age, we detect 21 large inversion polymorphisms segregating within deer mice (some of which are likely to overlap with rearrangements detected by cytogenetics). In localizing these inversions, we determine their positions relative to centromeres and telomeres, explore their effects on chromosome structure, characterize genomic content at their breakpoints and propose a mechanism by which inversions arise in this species. Further, we quantify the impact of the inversions on recombination and the resulting effects on mutational load. Finally, we survey the distributions of the inversions across the species range and identify several inversions that contribute to local adaptation. Taken together, these results reveal proximate and ultimate mechanisms involved in the establishment and maintenance of inversion polymorphisms and suggest a prominent role for these inversions in local adaptation.

## Results

### Identifying inversion polymorphisms

To identify putative inversion polymorphisms, we initially focused on five populations—four deer mouse (*P. maniculatus*) and one oldfield

mouse (*Peromyscus polionotus*), which is nested within the *P. maniculatus* clade (Fig. 1a)—and performed whole-genome resequencing (15× coverage with Illumina short-read data) on 15 individuals per

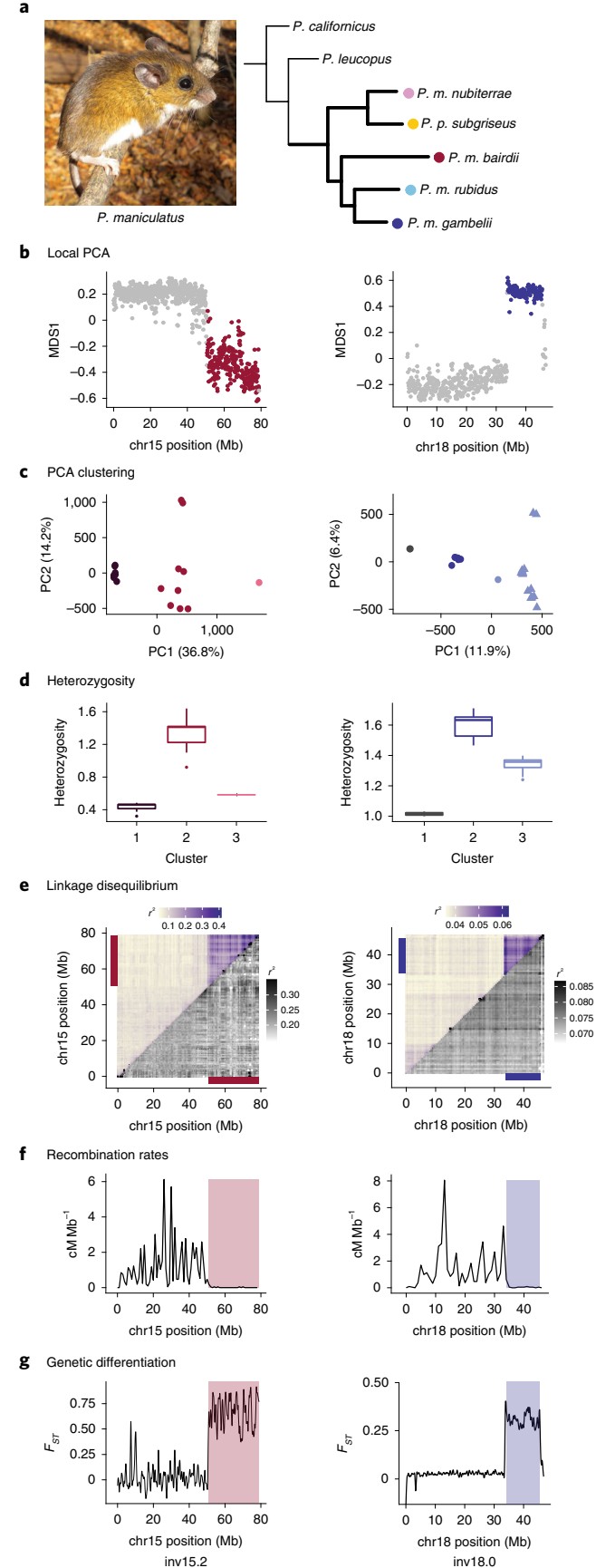

**Fig. 1 | Identifying inversion polymorphisms. a**, Left: photograph of *P. maniculatus* (Photo credit: E. P. Kingsley, reproduced from ref. [52], Wiley). Right: phylogenetic tree showing the relationships of five focal populations of *P. maniculatus* (bold branches) and two additional species. Note: *P. polionotus subgriseus* falls within the *maniculatus* clade. **b**–**g**, Detection of polymorphic inversions. **b**, Local PCA for example inversions in *P. m. bairdii* (left) and *P. m. gambelii* (right), where each dot represents a 100 kb window. Distances between local PCA maps are represented by the MDS1 axis, with outlier windows highlighted in colour (red or blue). **c**, Clustering of samples by PCA for entire outlier regions found in **b**, assigned using *k*-means clustering. Right: *P. m. rubidus* shown (triangles) for comparison. **d**, Heterozygosity of samples by cluster assignments from PCA in **c**. Boxes indicate upper and lower quartiles; centre line represents median; whiskers extend to minimum and maximum values within 1.5× interquartile range; points show outliers beyond whiskers. Sample sizes for clusters 1, 2, 3: (left) *n* = 7, 9, 1; (right) *n* = 2, 12, 16. **e**, LD for chromosomes (chr) harbouring the example inversions, shown as mean $r^2$ values for paired windows across each chromosome. Mean $r^2$ values including all samples from PCA clustering (upper triangle) and for only the more common homozygote genotype as determined by PCA clustering (lower triangle). Coloured bars highlight outlier regions from **b**. Scales for $r^2$ values are provided. **f**, Recombination rates (cM Mb⁻¹) shown for laboratory-born inversion heterozygotes. Outlier regions found in **b** are highlighted. **g**, $F_{ST}$ between homozygous genotypes (clusters 1 and 3 from **c**). Outlier regions found in **b** are highlighted.

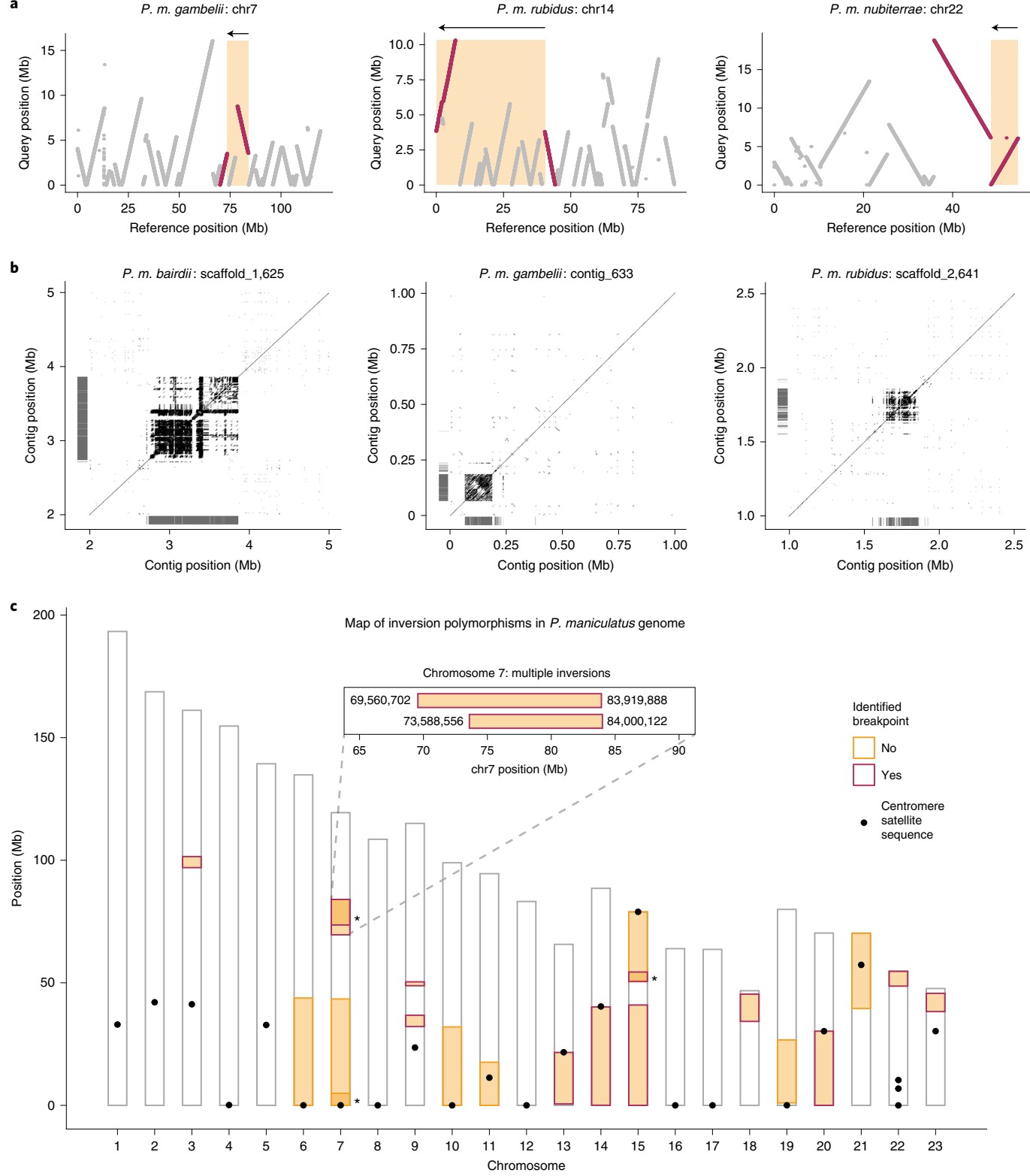

**Fig. 2 | Genome-wide map of inversions. a**, Three examples of contigs highlighting inversion breakpoints. Contigs from de novo genome assemblies ('query', *y* axis) were aligned to the *P. maniculatus* reference genome ('reference', *x* axis) with nucmer. Contigs (grey) and those identifying inversion breakpoints (red) are shown. Predicted inversion boundaries are highlighted (orange box), showing predicted inversion (arrow) above. **b**, Three examples of predicted centromeres in de novo genome assemblies. Dotplots show self-versus-self alignments with alignment length >100 bp. Locations of centromere satellite sequence alignments are shown (grey lines). **c**, Locations of inversions (*n* = 21) across chromosomes, with predicted centromeres (black dots). Asterisks highlight overlapping inversions on chromosomes 7 and 15; the inset shows the positions of identified breakpoints for two overlapping inversions on chromosome 7.

population. To identify patterns of genetic variation consistent with inversion polymorphisms, we first characterized local population structure within populations and between population pairs in 100 kb windows across the genome using local principal component analyses (PCA)[25] and identified outlier regions (Fig. 1b and Extended Data Fig. 1; as described in refs. [16,26]). We then focused on genomic regions for which the first principal component separated individuals into three clusters, probably representing the three possible inversion genotypes (Fig. 1c and Extended Data Fig. 1), with the central cluster having the highest heterozygosity, consistent with inversion heterozygotes (Fig. 1d and Extended Data Fig. 1).

To verify that these genomic patterns were driven by suppression of recombination between haplotypes, we next measured linkage disequilibrium (LD) and recombination rates. In wild-caught mice, LD across all genotypes (but not within homozygotes) was elevated within predicted inversion regions (Fig. 1e and Extended Data Fig. 1), suggesting that recombination is suppressed between but not within haplotypes. We also estimated recombination rates using laboratory-raised inversion heterozygotes and found that putative inversion regions showed nearly complete suppression of recombination in heterozygotes (mean recombination per inversion: 0–0.03 cM Mb⁻¹; Fig. 1f and Extended Data Fig. 1). Together, these results suggest that suppression of recombination is specifically driven by heterozygotes, providing strong evidence that inversion polymorphisms occur in the identified regions. In total, using this approach, we identified 21 inversion polymorphisms in this species. This is a conservative estimate because our approach was limited to identifying inversions >1 Mb in length with a minimum allele frequency of ~10%.

Owing to their number and sizes, these inversions alone affect recombination rates on a massive scale. The detected inversions range in size from 1.5 to 43.8 Mb and, in total, span 17.5% of the deer mouse genome. These inversions cause a near-complete suppression of recombination in heterozygotes: inversion regions show an average recombination rate of only 0.01 cM Mb⁻¹, compared with a genome-wide rate (excluding inversion regions) of 0.80 cM Mb⁻¹ (Extended Data Fig. 2). We also found no significant correlation between inversion size and recombination rate, highlighting how even the largest inversions almost completely suppress recombination (Extended Data Fig. 2). As a consequence, inversions can trap existing mutations or accumulate new mutations and maintain them in LD. Indeed, we found that genetic differentiation ($F_{ST}$) between inversion and standard haplotypes was elevated in a block-like structure (Fig. 1g and Extended Data Fig. 1), suggesting that the inversions partition genetic variation into large haploblocks, shaping patterns of genetic diversity across the deer mouse genome.

**Inversion breakpoints**

To localize inversion breakpoints, we performed PacBio long-read sequencing for one individual from each of the five focal populations and created de novo genome assemblies at the contig level (Extended Data Table 1). By aligning the de novo genome assemblies to the deer mouse reference genome (NCBI accession: GCA_003704035.3), we identified breakpoints for 13 of the 21 inversions (Fig. 2a and Extended Data Fig. 3). The eight inversions for which we did not identify breakpoints included five inversions (inv6.0, inv7.0, inv7.1, inv19.0, inv21.0) not represented in homozygotes among the PacBio-sequenced individuals (Extended Data Table 2); repetitive sequences probably prevented assembly across breakpoints for the remaining three inversions (inv10.0, inv11.0, inv15.1). Using the de novo genome assemblies, we predicted unique centromere locations for 21 of the 23 autosomes using a 344 bp satellite sequence that localizes to deer mouse centromeres[27]. Although centromeres are notoriously difficult to assemble[28], the de novo genome assemblies spanned multiple predicted centromeres, revealing the highly repetitive nature of centromeric regions, with satellite sequence repeats spanning as

much as 1.1 Mb (Fig. 2b). Together these data allowed us to precisely map many of the inversions to chromosomes and their positions relative to centromeres.

We found that the distribution of the inversion polymorphisms across the genome is nonrandom. Of the 21 inversions, 15 are terminal, where the inversion ends within 1.5 Mb of the end of the chromosome (Fig. 2c). In addition, nine inversions have breakpoints (predicted or identified) within 1 Mb of the centromere (Fig. 2c); as predicted centromeres localize within the three inversions with identified breakpoints (inv13.0, inv14.0, inv20.0) and the other six inversions (inv6.0, inv7.0, inv7.1, inv10.0, inv15.1, inv19.0) are terminal and occur on acrocentric chromosomes, these inversions are likely to be pericentric (contain the centromere). As such, these nine inversions may toggle chromosomes between acrocentric and metacentric states, shifting centromere locations by as much as 43 Mb. In addition, these results suggest that centromeric and telomeric regions are likely to harbour inversion breakpoints in deer mice.

We also identified multiple genomic regions with recurrent inversion breakpoints. For example, on chromosome 7, we detected two overlapping inversions (inv7.2, inv7.3) with nearly identical breakpoints localizing only 80.2 kb apart (Fig. 2c, inset). Using whole-genome alignments between *P. maniculatus* and *Peromyscus californicus*, an outgroup, we determined the ancestral versus derived orientation for these two inversions and found that they arose independently rather than as a series of nested inversions. We also identified two inversions on chromosome 15 (inv15.1, inv15.2) with a shared breakpoint (although we localized breakpoints for only one of these inversions with the de novo assemblies) and two additional inversions on chromosome 7 (inv7.0, inv7.1) with breakpoints both occurring near the telomere (although we were unable to localize breakpoints for either) (Fig. 2c). The recurrence of inversion breakpoints further suggests that certain genomic regions have a greater tendency to participate in the formation of chromosomal rearrangements.

Characterizing the nature of inversion breakpoint regions is critical to understanding how inversions arise and why some genomic regions may be more susceptible to breakpoints. There are two major mechanisms by which inversions form: (1) nonhomologous end joining (NHEJ) can create inversions if double-stranded breaks occur and the sequence is reintegrated in reverse orientation; and (2) nonallelic homologous recombination (NAHR) can yield inversions if intrachromosomal crossing over occurs between inverted repeats (Fig. 3a). For 12 of the 13 inversions with localized breakpoints, we identified at least one pair of inverted repeats flanking the inversion (Fig. 3b). These inverted repeats ranged from 500 bp to 50 kb in length (Fig. 3b) and were often duplicated near the breakpoints (Fig. 3c and Extended Data Fig. 4). This suggests that the vast majority of inversions for which we identified breakpoints likely arose owing to NAHR, as opposed to NHEJ, consistent with the formation of inversions in humans[29].

We next explored whether the breakpoints were enriched in repetitive genomic regions. For the 20 localized inversion breakpoints (excluding six breakpoints at chromosome ends), we used SEDEF[30] to identify segmental duplications (SDs), defined as duplicated sequence within 500 kb of the breakpoint that is >1 kb in length and contains <70% common repeats. We found that breakpoint regions were significantly enriched for SDs compared with randomized regions genome-wide (Kolmogorov–Smirnov test: $P < 0.001$); for example, 50% of breakpoints had SD density in the top 90th percentile of random regions genome-wide (Fig. 3d). The repetitive structure of the breakpoints varied, with some breakpoint regions harbouring highly structured SDs in tandem (Fig. 3e and Extended Data Fig. 4) and others harbouring multiple interspersed SDs (Extended Data Fig. 4). Together, these analyses show that genomic regions with an accumulation of SDs may be prone to chromosomal rearrangements via ectopic recombination in deer mice.

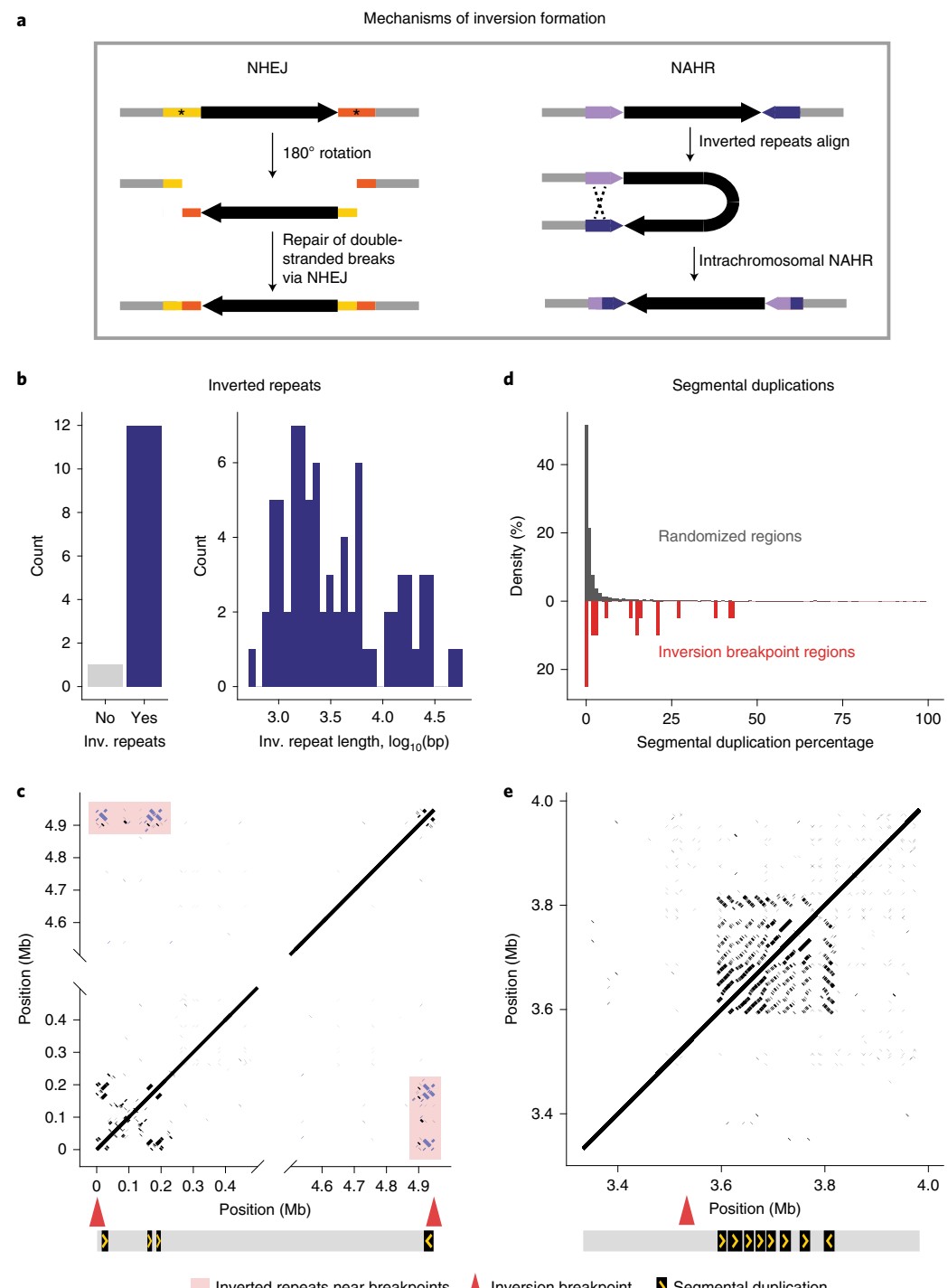

**Fig. 3 | Inversion breakpoints. a**, Two primary mechanisms by which chromosomal inversions form. Left: inversions form when double-stranded breaks (asterisks) occur and sequence is reincorporated via NHEJ in an inverted orientation. Right: inversions form via NAHR between inverted repeats (blue and purple arrows) on the same chromosome. **b**, Numbers of inversions for which inverted repeats are present or absent within 500 kb of both inversion breakpoints; histogram shows the distribution of lengths of identified inverted repeats. **c**, Example of an inversion on chromosome 9 with inverted repeats. Dotplot shows self-versus-self alignments for a long-read assembly contig spanning the inv9.0 inversion. Locations of breakpoints (red arrows) are shown; only alignments with length >100 bp and within 500 kb of the breakpoints are shown. Inverted repeats mapping to within 500 kb of both breakpoints are shown (purple) and highlighted (pink box). Diagram below shows position and orientation of inverted repeats. **d**, Percentage of sequence assigned as SDs in randomly selected 1 Mb regions across the genome ($n = 10,000$, grey) or within 500 kb of inversion breakpoints ($n = 20$, red). SDs are defined as regions >1 kb that are duplicated within the selected region, with >70% identity and <70% of sequence masked as common repeats. **e**, Example of a region harbouring SDs near an inversion breakpoint on chromosome 7 (inv7.3). Dotplot shows self-versus-self alignments with length >100 bp for a long-read assembly contig spanning the inv7.3 breakpoint; the location of the breakpoint (red arrow) is shown. The diagram below shows the position and orientation of a 15-kb region duplicated eight times. Inv., inverted.

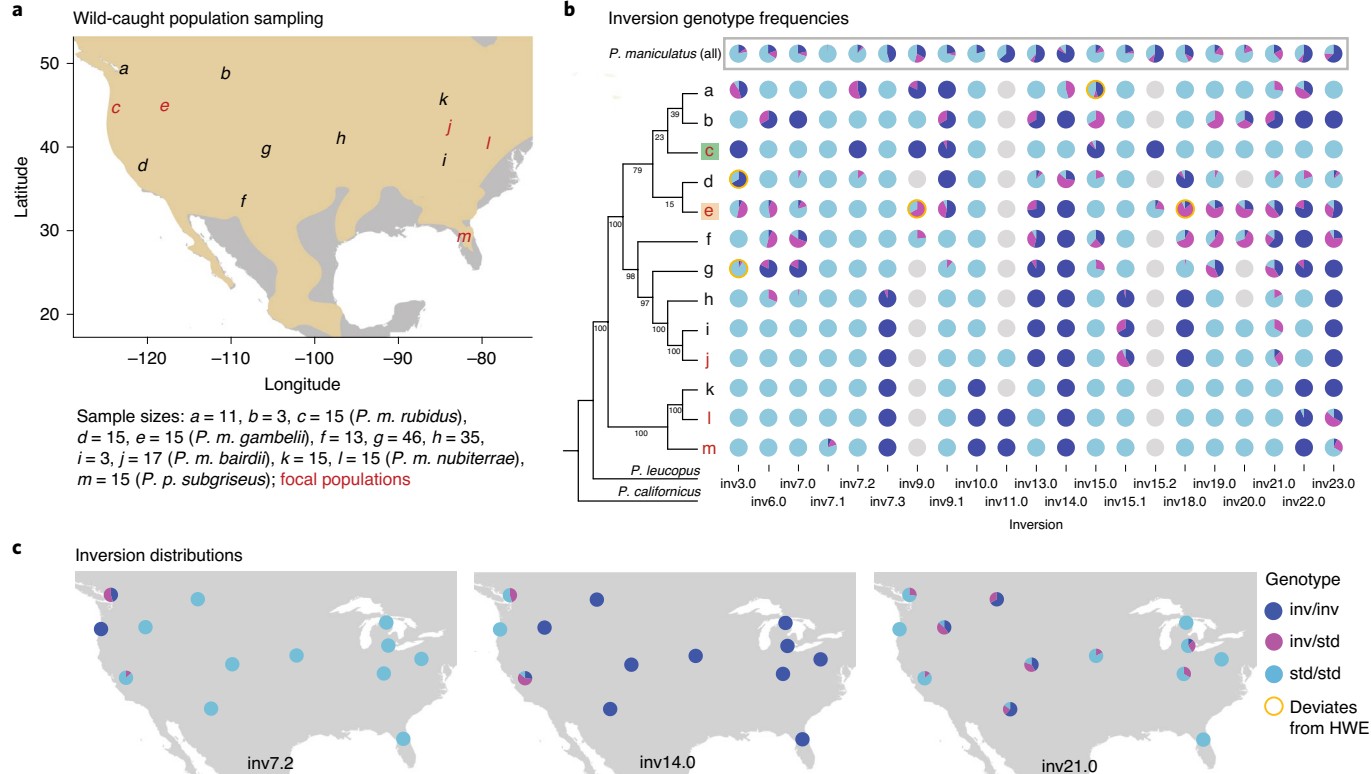

**Fig. 4 | Distributions and frequencies of inversions. a**, Locations of all *P. maniculatus* populations sampled (*n* = 13), labelled *a–m*; sample sizes given below. The five focal populations are highlighted (red). The approximate *P. maniculatus* range is highlighted in tan. **b**, Inversion genotype frequencies for each population: letters correspond to populations in **a**. Cladogram (left) shows relatedness (excluding inversion regions) between populations and two outgroups (*P. leucopus*, *P. californicus*); bootstrap values are shown. Inversion genotype frequencies across all *P. maniculatus* populations are shown at the top (grey box). Populations that deviate from HWE equilibrium are shown (*n* = 5, yellow outline). Missing data are shown as grey circles. Green and tan boxes highlight forest and prairie ecotypes, respectively. **c**, Distributions of three example inversions (inv7.2, inv14.0 and inv21.0) are shown, with inversion genotype frequencies for each population. Additional inversion distributions are provided in Extended Data Fig. 5. inv, inverted haplotype; std, standard haplotype.

## Frequencies and evolution of inversions

To explore the distributions of these inversions, we next characterized their frequencies across the species range. We first determined the derived inversion arrangement based on genome alignments with an outgroup, *P. californicus*, and then genotyped the inversions in 218 mice from 13 populations (Fig. 4a). Most inversions were found in multiple populations: 18 of the 21 inversions were present in at least three of the 13 sampled populations (Fig. 4b). However, the varying distributions of the inversions suggest that they have differing evolutionary histories (for example, inversion age and selection): some inversions (for example, inv14.0) are widespread, whereas others (for example, inv7.2) are spatially constrained (Fig. 4c and Extended Data Fig. 5). The highly polymorphic nature of many of the inversions (for example, inv21.0) (Fig. 4c and Extended Data Fig. 5) was particularly striking, with 16 of 21 inversions segregating in at least two of the sampled populations (Fig. 4b). As such, inversion heterozygotes are common (Fig. 4b), indicating that the inversions have a profound impact on recombination rates in the wild.

**Limited evidence for deleterious effects of inversions.** To explore any negative consequences of inversions on fitness, we first examined possible deleterious effects due to inversion breakpoints. If an inversion breakpoint occurs within or near a gene, it may substantially affect the function and/or expression of that gene[31]. We found that significantly fewer inversion breakpoints occurred within protein-coding genes than expected based on the deer mouse gene density (binomial test: *P* = 0.004): of the 13 inversions for which we localized breakpoints,

only two inversions (inv9.1, inv18.0) had breakpoints occurring within a protein-coding gene (inv9.1 disrupts the *1700129C05Rik* intron, inv18.0 disrupts the *Slc39a5* coding sequence (left breakpoint) and *Baz2a* intron (right breakpoint)) (Fig. 5a). Whereas these two inversions may affect phenotypes through disrupting gene function, the other 11 inversions with localized breakpoints do not disrupt annotated genes (Fig. 5a) and are thus less likely to convey strongly deleterious effects, although their breakpoints may still influence gene expression.

We next characterized possible mutational loads carried by the inversions, which may accumulate owing to suppressed recombination in inversion heterozygotes[32]. To do so, we tested whether the inversions were enriched for nonsynonymous mutations relative to the standard haplotypes. We found that the inversions did not show a significant increase in their proportion of segregating nonsynonymous to synonymous mutations ($pN/pS$) compared with the standard haplotypes (two-sided *t*-test: *P* > 0.05 for all inversions), nor did they show a significant increase in nucleotide diversity at nonsynonymous versus synonymous sites ($\pi_N/\pi_S$) compared with the standard haplotypes (two-sided *t*-test: *P* > 0.05 for all inversions) (Fig. 5b). In addition, neither the inversions nor the standard haplotypes showed enrichment for nonsynonymous mutations ($pN/pS$ and $\pi_N/\pi_S$) relative to the rest of the genome (one-sided *t*-test: *P* > 0.05 for all inversions and standard haplotypes) (Fig. 5b), and we did not find a correlation between inversion heterozygote frequency and mutational load (Extended Data Fig. 6). Using nonsynonymous mutation accumulation as an estimate of mutational load, these results suggest that the inversions do not harbour a strong deleterious mutational load.

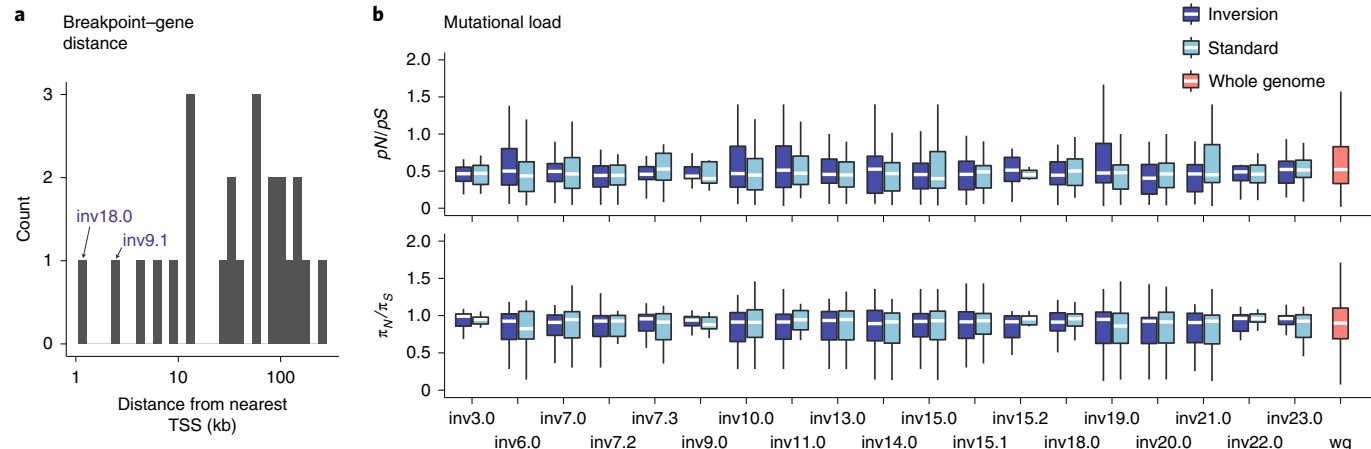

**Fig. 5 | Limited evidence for deleterious effects of inversions. a**, Distances between inversion breakpoints (for 13 inversions with localized breakpoints) and the nearest transcriptional start site (TSS). Two inversions (inv18.0, inv9.1) with breakpoints disrupting protein-coding genes are highlighted. **b**, $pN/pS$ and $\pi_N/\pi_S$ computed in 500 kb windows, shown for inversion versus standard haplotypes and whole-genome (wg) regions excluding the inversions. Number of windows: $n = 8$–$42$ (inversions); $n = 2{,}725$ (whole genome). Boxes indicate upper and lower quartiles; centre line represents median; whiskers extend to minimum and maximum values within 1.5× interquartile range. Differences between inversion and standard haplotypes in $pN/pS$ and $\pi_N/\pi_S$ are all nonsignificant (two-sided $t$-test: $P > 0.05$), and neither inversions nor standard haplotypes are enriched for $pN/pS$ and $\pi_N/\pi_S$ compared with the whole genome (one-sided $t$-test: $P > 0.05$).

In addition, if inversions accumulate a recessive mutational load, inversion homozygotes should be rare (for example, in butterflies[33] and sparrows[34]). In deer mice, however, inversion genotype frequencies are consistent with Hardy–Weinberg equilibrium (HWE): we found only five (of 73) instances in which a segregating inversion significantly deviated from HWE within a population (Fig. 4b). This suggests that inversion homozygotes are not strongly underrepresented relative to expectation in populations segregating for a given inversion, which further supports the observation of limited mutational load. We also note that since most inversion genotype frequencies are consistent with random mating, strong assortative or disassortative mating by inversion genotype does not readily occur (unlike in the ruff[6,7] or white-throated sparrow[34]). Together, these lines of evidence suggest that these inversions in deer mice are not associated with strongly negative effects on fitness.

**Multiple inversions contribute to local adaptation.** To explore the role of positive selection in the establishment and maintenance of these inversion polymorphisms, we characterized the contribution of inversions to local population differentiation. We took advantage of previous work on two populations, representing forest and prairie deer mouse ecotypes (populations *c* and *e*, Fig. 4a), which are well characterized and widespread[20]. Forest and prairie mice show many pronounced phenotypic differences (for example, coat colour, tail length, foot length) despite ongoing gene flow. We previously identified an inversion on chromosome 15 (inv15.0) that contributes to phenotypic divergence between these ecotypes[20]. Returning to this system, we found that multiple newly identified inversions were also major contributors to differentiation between these populations. Specifically, genome-wide $F_{ST}$ is low between ecotypes (genome-wide forest–prairie $F_{ST}$: $0.03 \pm 0.03$) owing to high migration rates[20], yet we found multiple 'genomic islands of divergence' that showed remarkable overlap with identified inversion polymorphisms (inversion-region forest–prairie $F_{ST}$: $0.26 \pm 0.16$) (Fig. 6a). For 13 inversions, the ecotypes differed by >50% in their inversion frequencies. Using forward-genetic simulations in SLiM[35], we found that for a locus to be maintained at >50% frequency difference between the forest and prairie ecotypes given high gene flow, it was most likely to be evolving under divergent selection (Extended Data Fig. 7), implicating these 13 inversions in local adaptation.

The distributions of these inversions across a forest–prairie habitat gradient further support their role in adaptation. Specifically, we genotyped the 13 polymorphic inversions in 136 samples across an environmental gradient and found that nine inversions showed steep changes in frequency across the forest–prairie habitat transition (Fig. 6b and Extended Data Fig. 8), suggesting that these inversions may be favoured in alternate habitats. Furthermore, five inversions (inv7.2, inv14.0, inv15.0, inv18.0, inv21.0) were significantly associated with an ecotype-defining trait, tail length, in laboratory-raised $F_2$ hybrids[20] ($P < 0.05$, linear model) and, for all five, the forest arrangement was associated with longer tails (Fig. 6c), consistent with long tails being important for balance in arboreal habitats[36]. These five inversions together explain 23.0% of the variance in tail length (individually explaining 2.0–12.5% of the variance, with additive effects ranging from 1.1–2.7 mm change in tail length). Inv15.0 has also been previously found to be significantly associated with coat colour, a second ecotype-defining trait[20] (explaining 40% of coat colour variance) (Fig. 6c). Together, these results suggest that inversions may be a key source of genetic variation differentiating locally adapted deer mouse populations, with divergent selection likely to play a role in maintaining the inversions as polymorphisms within this species.

## Discussion

Technological advances in genome sequencing have recently led to new opportunities for characterizing intraspecific structural variation. For example, the ability to perform population-level whole-genome resequencing allows signatures of large structural variants such as chromosomal inversions to be more easily detected[19]. This approach has recently been successful in identifying inversions in sunflowers[16,26] and seaweed flies[15] and now in deer mice. In addition, long-read sequencing has also greatly facilitated the detection and classification of structural variants. For example, here we found that inversion breakpoints reside in highly repetitive genomic regions, harbouring an enrichment of segmental duplications, similar to other mammalian species (that is, humans and great apes[13,17]). The repetitive nature of mammalian inversion breakpoints probably explains why breakpoints are so challenging to detect with short-read sequencing data alone, as well as with long-read data if read length or coverage is insufficient to resolve repeat regions, as we suspect is the case for the deer mouse inversions for which we failed to localize breakpoints. Future work combining these two approaches—to perform population-level long-read genome sequencing—will further our ability to detect structural variation within a diversity of species[18].

In discovering deer mouse inversion polymorphisms, we found that they have an interesting distribution in the genome: a majority of

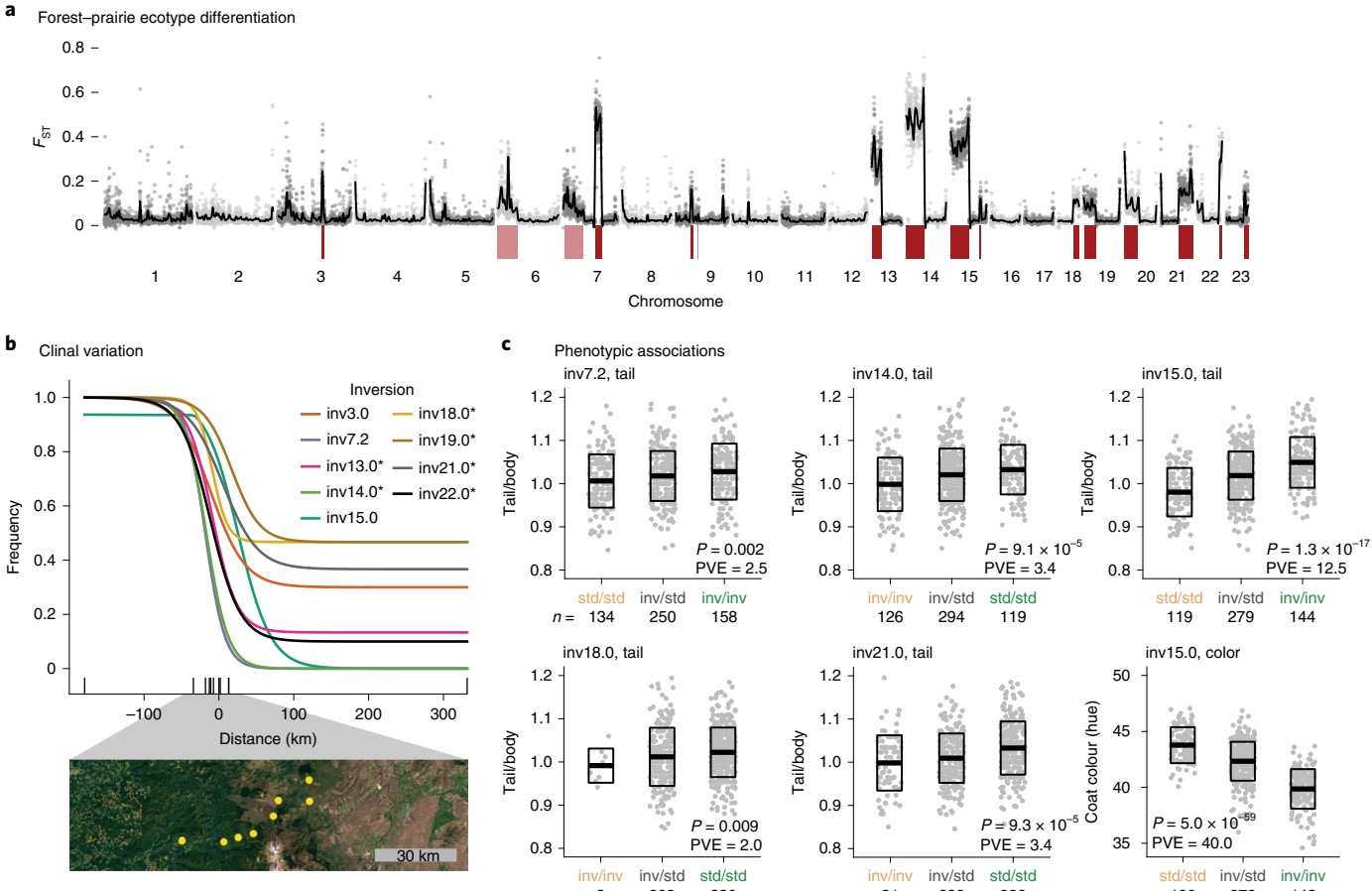

**Fig. 6 | Inversions involved in local adaptation. a**, $F_{ST}$ between forest and prairie ecotypes found in western North America. Grey dots show $F_{ST}$ from 100 kb windows, with smoothed $F_{ST}$ as black lines. Inversions with allele frequency difference >50% between forest and prairie ecotypes are shown in dark red; inversions with allele frequency difference >10% are shown in pink. **b**, Inversion or standard arrangement frequencies across an environmental gradient found between the forest and prairie populations, with clines fit using hzar. Sampled populations are shown with black ticks along the *x* axis, with forest population as the left-most tick and prairie population as the right-most tick. Clines show frequency of the forest allele: an asterisk indicates that the standard arrangement is more common in the forest population; no asterisk indicates that the inversion

arrangement is more common in the forest population. A satellite image across the environmental gradient (obtained from GoogleEarth) is shown below, with sampled sites in yellow. **c**, Inversions with significant associations with tail length or coat colour in a forest–prairie $F_2$ intercross. $F_2$ hybrids are grouped by inversion genotype, with each genotype coloured by whether it is more common in the prairie (tan) or forest (green) population. Sample sizes are shown below the *x* axes. Data represent means (centre lines) ± s.d. (boxes). *P* values (Bonferroni corrected) and percentage variance explained (PVE) from linear models, calculated with analysis of variance, are shown for each association. Tail/body, ratio of tail length to body length; coat colour is measured by hue and shown in degrees. inv, inverted haplotype; std, standard haplotype.

the inversions occur terminally, and most of these involve breakpoints near centromeres. The inversions with breakpoints adjacent to centromeres are likely to be shifting centromere locations from the middle of the chromosome to the end of the chromosome (and vice versa), transforming chromosomes between metacentric and acrocentric states. This result could explain the longstanding observation that deer mice vary in number of acrocentric chromosomes[23,24]. Furthermore, inversions are also likely to influence chromosome accessibility owing to changes in the three-dimensional genome structure, which, in addition to the mutations the inversions carry, may influence the expression of genes found within the inversions. Despite this large variation in chromosome structure, deer mice (and, more generally, the *Peromyscus* genus) have a strongly conserved chromosome number (diploid $n = 48$)[24]. Unlike the case in other rodents such as the house mouse, which harbours Robertsonian fusions[37], the large rearrangements involving centromeres occur primarily within and not between chromosomes in deer mice.

One hypothesis for why deer mouse inversions tend to involve telomeric and centromeric regions is that inversion breakpoints arise more frequently in these regions: genomic regions near centromeres

and telomeres can harbour an excess of SDs (as well as other repeats), which may facilitate ectopic recombination[38]. A second hypothesis is that inversions with breakpoints in telomeric or centromeric regions are less likely to be removed by purifying selection than inversions that occur in other genomic regions: breakpoints that occur near centromeres and telomeres may be unlikely to have deleterious effects as these regions tend to be gene-sparse[38]. Indeed, none of the inversion breakpoints we found near centromeres (and only one near a telomere) disrupted protein-coding sequences. Terminal inversions may also be less likely than non-terminal inversions to have strong underdominant effects, which often occur owing to inversion loops that form in heterozygotes during meiosis[3]. If an inversion lacks homologous sequence on one side, such as in a terminal inversion, loop formation may be prevented. Previous evidence from deer mice suggests that inversion loop formation is rare in putative terminal inversions[39]. Thus, deer mouse inversions involving telomeres and centromeres may confer fewer deleterious costs associated with breakpoint effects and underdominance than inversions occurring in the rest of the genome.

Inversions are a particularly interesting form of structural variation because of their effects on recombination. Inversions in deer

mice, when heterozygous, suppress recombination across their entire lengths. The number and sizes of the inversions thus seem striking in the context of recombination: 21 detected inversion polymorphisms, with a mean length of 20.0 Mb, affect a total of 420 million DNA bp (or 17%) of the deer mouse genome. Although these results are consistent with large inversions causing suppression of recombination in other species (for example, quails[40], maize[41] and cod[42]), whether inversion polymorphisms affect similar proportions of the genome in other species remains largely unknown. Furthermore, as our approach was limited to detecting inversions >1 Mb in length, there are possibly many additional inversions of shorter lengths segregating within deer mice, which is an important direction for future work. Nevertheless, we found that the detected inversions substantially shape the recombination landscape of deer mice: although suppression of recombination is limited to inversion heterozygotes (so the frequency of an inversion will determine the extent to which it affects recombination), most deer mouse inversions are widespread and inversion heterozygotes are common in natural populations.

Recombination plays an important role in evolution through creating new combinations of alleles and increasing the efficiency of natural selection[43]. In particular, through uncoupling deleterious and beneficial mutations, recombination reduces Hill–Robertson interference and facilitates the elimination of deleterious mutations and the spread of beneficial mutations[44,45]. Given the benefits of recombination, the abundance of inversions presents a paradox. With reduced efficacy of purifying selection in the absence of recombination, the expectation is that inversions will accumulate a deleterious mutational load (when inversion heterozygotes are common)[32], which will limit their spread[46]. In deer mouse inversions, however, we did not find evidence for the accumulation of mutational load based on nonsynonymous mutations (although these inversions may harbour an excess of other types of deleterious variants such as transposable elements, which future work will further resolve), consistent with a recent study in sunflowers[47]. In both deer mice and sunflowers, inversion homozygotes are common[47]; as recombination proceeds uninterrupted in inversion homozygotes, deleterious mutations can efficiently be removed once an inversion reaches substantial allele frequency[32], especially if effective population sizes ($N_e$) are high, as in many populations of deer mice (for example, $N_e \approx 4 \times 10^6$ in a single population[20]). As in sunflowers[47], we hypothesize that these inversions, which act as large-scale modifiers of recombination when heterozygous, largely evaded deleterious costs associated with suppressed recombination by quickly spreading to high frequencies in deer mice, whose large population sizes could facilitate effective purifying selection in inversion homozygotes[32] (noting that gene conversion between inversion and standard haplotypes may also have a role in reducing deleterious mutational load[32]).

A major hypothesis for the maintenance of inversion polymorphisms is the 'local adaptation hypothesis', which posits that when a population is locally adapting in the face of gene flow, suppressed recombination between multiple beneficial mutations can be advantageous, reducing the strength of selection necessary to establish and/or maintain each mutation in migration–selection equilibrium[9–11]. As deer mice are found continuously across a wide range of habitats, they are subjected to a range of selective pressures, probably with ongoing gene flow. Our results support an important role for divergent selection in maintaining inversions as polymorphisms within the species at large. In particular, we found that 13 inversions, including one previously identified[20], are segregating between forest and prairie deer mouse ecotypes with high allele frequency differences and are likely to be subject to habitat-associated divergent selection, consistent with multiple inversions differentiating ecotypes in a diversity of species such as snails[48], cod[42], sunflowers[26] and sticklebacks[49]. Although it remains an open question whether the inversions segregating between these forest–prairie ecotypes are advantageous because of their suppression of recombination, the high levels of migration between the forest and prairie populations suggest that increased LD between adaptive alleles may be particularly beneficial in this system[20]. In addition, five of these inversions have significant effects on tail length, and thus variation in this ecotype-specific trait is largely partitioned into inversions, consistent with the evolution of concentrated genetic architectures in the face of gene flow[50].

A concrete understanding of the prevalence and significance of inversion polymorphisms specifically, and of structural variation more generally, remains largely elusive across natural populations of organisms, especially mammals[51]. We find that inversion polymorphisms are abundant in deer mice. Whether the abundance of inversion polymorphisms in deer mice is unique or representative of mammalian species will require similar investigations across additional species. Nevertheless, this work highlights the critical role of inversions in shaping patterns of recombination, genetic diversity and chromosomal structure in the deer mouse and suggests that inversions may play an even more important part in local adaptation than previously appreciated.

## Methods

### Population sampling and sequencing

**Focal population sampling.** We focused our initial analyses on five populations of *P. maniculatus*, each representing a distinct subspecies (*P. m. rubidus*, *P. m. gambelii*, *P. m. bairdii*, *P. m. nubiterrae* and *P. p. subgriseus*). Tissues from 15–17 wild-caught mice per population were collected in Siuslaw National Forest, Oregon, USA (*P. m. rubidus*)[20], Baker City, Oregon, USA (*P. m. gambelii*)[20], Derry, Pennsylvania, USA (*P. m. nubiterrae*)[52], Ocala National Forest, Florida, USA (*P. p. subgriseus*)[53] and Bridgewater, Michigan, USA (*P. m. bairdii*; obtained from the University of Michigan). All samples used in this study are listed in Supplementary Table 1.

**Whole-genome resequencing and variant calling.** To generate whole-genome resequencing data, we first extracted DNA from ~20 mg of liver tissue and generated sequencing libraries using Illumina DNA library preparation kits. We sequenced the resulting libraries using 150 bp paired-end sequencing on an Illumina NovaSeq S4 flow cell to obtain ~15× coverage per sample. Following demultiplexing, we mapped sequencing reads to the *P. maniculatus bairdii* reference genome (NCBI accession: GCA_003704035.3) using BWA-MEM. We accessed published whole-genome resequencing data for three populations: *P. m. rubidus*, *P. m. gambelii*[20] (NCBI: PRJNA688305) and *P. p. subgriseus*[53] (PRJNA838595). To call variant sites, we used HaplotypeCaller (GATK3.8) on each sample with the default heterozygosity prior (−hets = 0.001) and −ERC GVCF to produce per-sample genomic variant call format files (vcfs). Then, we ran GenotypeGVCFs (GATK3.8) to jointly genotype the samples. We performed hard filtering of SNPs based on GATK best practices (filtering variants with quality by depth (QD) < 2.0, FisherStrand (FS) > 60.0, mapping quality (MQ) < 40.0, MQRankSum < −12.5, ReadPosRankSum < −8.0) using VariantFiltration.

### Identifying inversions

**Local PCA.** To identify genomic regions with outlier population structure, we performed local PCA with the lostruct package[25] in R on each of the five focal populations and for all focal population pairs. Note that when all populations are included, population structure is driven by population divergence, which masks the signatures of possible inversions. Therefore, we included only individual populations or population pairs for this analysis, such that inversion signatures were detectable. Using lostruct, we performed local PCA for 100 kb windows with a step size of 100 kb. We then computed the distance between PCA maps (with the top two PCs) using the pc_dist function with default parameters and visualized these distances using multidimensional scaling (MDS) with the cmdscale function with two MDS axes.

To identify genomic regions with unusual population structure, we scanned for consecutive 100 kb windows that showed similar

population structure to each other and distinct population structure from the rest of the chromosome. To do so, we first performed $k$-means clustering of the 100 kb windows in the MDS space, defined by the MDS1 and MDS2 axes, using numbers of clusters from $k = 2$ to $k = 10$. To determine the best $k$, we chose the $k$ with the maximum silhouette score; this is an averaged measure of the dissimilarity between an observation and its neighbouring cluster. We then assigned 100 kb windows to the cluster determined by the $k$-means clustering for the chosen $k$. We next calculated the $z$ score for the MDS1 score for each 100 kb window and selected genomic regions with consecutive windows belonging to the same cluster in which at least ten consecutive windows had $z$ score >1.5.

**PCA and heterozygosity.** For each identified outlier region, we performed PCA on the entire region using scikit-allel v.1.3.2 (https://github.com/cggh/scikit-allel). For scikit-allel analyses, we created zarr objects from the whole-genome resequenced vcfs using allel.vcf_to_zarr. We then performed PCA using all SNPs in the region, with the function allel.pca, with n_components = 10, scaler = 'patterson' and ploidy = 2. $k$-means clustering of samples in PC1 versus PC2 space was performed in R with kmeans, following the approach detailed by Todesco et al.[16], where samples were assigned to three clusters, setting the cluster starting positions as the minimum, maximum and middle value for PC1 scores to prevent clustering from being influenced by unequal numbers of samples per cluster. When clustering into three groups failed, we tried clustering into two groups, which would be the case if only two inversion genotypes are present. In a few cases ($n = 4$), we manually reassigned clusters for samples when $k$-means clustering had clear misassignments. For each outlier region identified, we also computed heterozygosity (reported as the percentage of sites that are heterozygous) for every sample in the relevant populations, using count_het in scikit-allel. Finally, we selected putative inversions to be outlier regions for which samples clustered into three distinct groups along PC1 with high heterozygosity for the middle cluster. We also included an additional four regions for which samples clustered into only two distinct groups along PC1 but signatures of recombination suggested the presence of an inversion (see below).

**Linkage disequilibrium.** For each putative inversion, we computed LD across the chromosome harbouring that putative inversion using: (1) all samples belonging to the population or population pair from which the putative inversion was identified; and (2) only the samples homozygous for the more common haplotype, based on the PCA clustering. To compute LD, we subset the vcf by sample and chromosome with bcftools. We then used vcftools to filter for SNPs with minor allele frequency (MAF) > 5% (--maf 0.05) and number of missing genotypes = 0 (--max-missing-count 0) and thinned SNPs to at most one SNP per 1 kb (--thin 1000). We computed LD with vcftools geno-r2. Finally, we used the script emerald2windowldcounts.pl (https://github.com/owensgl/reformat, https://github.com/owensgl/haploblocks) to calculate the mean $r^2$ between 500 kb windows (that is, for a given set of two 500 kb windows, the mean $r^2$ across all pairwise SNP comparisons between the two windows was computed).

**Recombination rates.** We estimated recombination maps for both the whole genome and within inversion regions, using laboratory-raised $F_2$ hybrids from previous intercrosses between two population pairs: *P. m. rubidus* × *P. m. gambelli*[20] and *P. m. bairdii* × *P. p. subgriseus*[54], which yielded a total of 547 and 1061 $F_2$ hybrids, respectively. Using double-digest restriction-site associated DNA (ddRAD) sequencing data of $F_2$ hybrids, we determined ancestry and the location of recombination breakpoints in the $F_2$ hybrids using the multiplexed shotgun genotyping pipeline (see ref. [20] for details). For the *P. m. rubidus* × *P. m. gambelli* intercross, we genotyped the founders ($n = 4$) and $F_1$ hybrids ($n = 49$) of the intercross for the inversions (see Genotyping samples for inversions) to ensure that only $F_2$ hybrids that were offspring of $F_1$

inversion heterozygotes were used for computing recombination rates within inversion regions. All inversions analyzed in the *P. m. bairdii* × *P. p. subgriseus* intercross were fixed between the founders. Five inversions (inv7.0, inv7.3, inv9.1, inv15.2, inv20.0) were not represented by heterozygous $F_1$ hybrids and so we were unable to characterize recombination rates for these inversions.

**Genetic differentiation.** To measure genetic differentiation between inversion and standard haplotypes across each identified inversion, we computed $F_{ST}$ between predicted homozygote genotypes (clusters 1 and 3 from PCA clustering) using scikit-allel. We performed sliding-window $F_{ST}$ analyses for 10 kb windows with a 10 kb step size using scikit-allel with the windowed_hudson_fst function and visualized $F_{ST}$ with loess smoothing in R.

To analyze genome-wide genetic differentiation between forest (*P. m. rubidus*) and prairie (*P. m. gambelii*) ecotypes, we computed $F_{ST}$ between forest and prairie populations in 100 kb windows across the genome with a step size of 100 kb, using scikit-allel with the windowed_hudson_fst function.

**Localizing inversion breakpoints**

**PacBio long-read sequencing and de novo genome assembly.** We performed long-read sequencing on five individuals (laboratory-colony-raised mice), one from each focal population. First, we extracted high-molecular-weight DNA from 200 µl fresh blood using the MagAttract HMW DNA mini kit (Qiagen), following the Whole Blood protocol. We quantified the resulting DNA using a Genomic DNA ScreenTape on a TapeStation 4200 (Agilent). Library preparation and sequencing were performed at the PacBio Sequencing Core of the University of Washington. In brief, libraries were prepared with the SMRTbell Express Template Prep Kit 2.0 (PacBio). We performed a size selection of 30 kb for the *P. m. rubidus*, *P. m. nubiterrae* and *P. m. bairdii* samples using BluePippin (Sage Science); we did not perform any size selection for the *P. m. gambelii* and *P. p. subgriseus* samples as the total library mass was below 500 ng. We then sequenced each on a Sequel II SMRTcell 8 M (PacBio), the *P. m. rubidus, P. m. nubiterrae* and *P. m. bairdii* samples with a 15 h video and the *P. m. gambelii* and *P. p. subgriseus* samples with a 30 h video.

We converted the bam files from each video to fastq files using bam2fastx (PacBio). We then used flye[55] to create de novo genome assemblies at the contig level for each population. The flye assembler uses a repeat graph to assemble across repetitive genomic regions, a critical feature for localizing inversion breakpoints, which often occur in repetitive genomic regions. To reduce run time, we downsampled to 40× coverage (-asm-coverage = 40) for initial disjointing assembly but otherwise ran the assembler with default parameters. Genome qualities are reported in Extended Data Table 1.

To genotype each PacBio sample for the inversions, we first mapped the PacBio fastq files to the *P. maniculatus* reference genome using ngmlr[56]. Then, we used longshot[57], a long-read-specific variant caller, to call variants for each sample. We merged the variant calls with the whole-genome resequencing vcfs and performed PCA for each inversion region, which allowed us to genotype the PacBio samples for the inversions (Extended Data Table 2; for details, see Genotyping samples for inversions).

**Inversion breakpoint identification.** We aligned the PacBio genome assemblies to the *P. maniculatus bairdii* reference genome using nucmer (mummer)[58] with default parameters. Owing to the possibility of reference genome errors, we reoriented any scaffolds in the reference genome that were misoriented relative to the *P. m. bairdii* long-read assembly (that is, we identified signatures of inversions or translocations in the *P. m. bairdii* long-read assembly relative to the reference genome and resolved these regions to match the *P. m. bairdii* long-read assembly). Thus, all inversion analyses were relative to the *P. m. bairdii*

long-read assembly. We also aligned published *P. californicus*[59] (NCBI accession: GCA_007827085.2) and *P. leucopus*[60] (NCBI accession: GCA_004664715.2) genomes as well as previously assembled de novo genomes for *P. m. rubidus* and *P. m. gambelii*[20] from canu (a long-read genome assembler complementary to flye) to the *P. maniculatus* reference genome using nucmer.

For each inversion, we scanned for evidence of inversion breakpoints. To do so, we filtered for nucmer alignments >4 kb in length (or >10 kb for *P. californicus*, *P. leucopus* alignments). Inversion breakpoints are identifiable if: (1) a contig spans the inversion region and maps to the reference genome in opposite orientation within the inversion region; or (2) a contig spans only part of the inversion region and maps to the reference genome in opposite orientation to the flanking region of the other end of the inversion. We thus identified contigs that showed signatures of inversions in predicted inversion regions and identified breakpoint positions based on the PacBio assembly alignments to the *P. maniculatus* reference genome. In addition, we identified breakpoints for one of the predicted inversions based on the *P. leucopus* genome alignment to the *P. maniculatus* reference genome and one of the predicted inversions based on the *P. californicus* genome alignment to the *P. maniculatus* reference genome.

**Determining derived arrangement.** For each inversion polymorphism, we determined which arrangement was ancestral (standard) versus derived (inversion) based on the whole-genome alignments between *P. californicus* (outgroup) and *P. maniculatus*. We evaluated whether the *P. californicus* reference genome was inverted relative to the *P. maniculatus* reference genome for each inversion region, and we assigned the *P. californicus* orientation to be the ancestral, standard arrangement.

**Predicting centromere locations.** *Peromyscus* species are known to have satellite sequences that map to centromeres; specifically, a 344 bp satellite sequence (NCBI accession: KX555281.1) localizes to *P. maniculatus* centromeres[27]. We used blastn (blast v.2.2.29) to map this satellite sequence to the *P. maniculatus* reference genome and to each PacBio genome assembly (as long-read genome assemblies are more likely to assemble across repetitive regions), filtering for alignments with >85% identity. Using this approach, we then determined centromere locations in the reference genome (converting alignment positions in the PacBio genomes to their corresponding or closest reference genome coordinates). To further explore the predicted centromeres, we created dotplots for contigs from the PacBio genomes that spanned a predicted centromere. To do so, we used nucmer with --maxmatch, -l 50, -c 100 to align each contig to itself and then plotted all alignments >100 bp using R.

**Characterizing repeat content at inversion breakpoints**
**Dotplots.** To evaluate whether inversion breakpoints occurred in repetitive regions, we created dotplots from the PacBio contig-level assemblies. We performed self-versus-self nucmer alignments for contigs spanning inversion breakpoints, with --maxmatch, -l 50, -c 100; we filtered for alignments >1 kb and plotted the alignments in R.

**Inverted repeats and SDs.** We identified inverted repeats and segmental duplications (SDs) near inversion breakpoints using the package SEDEF[30]. For the relevant PacBio contigs identified above (spanning or adjacent to inversion breakpoints), we softmasked common repeats with RepeatMasker, using --xsmall and --species rodentia and masked the 344 bp centromere satellite sequence. We then performed SEDEF with default parameters on the entire set of relevant PacBio contigs. First, we determined inverted repeats to be any repeat identified by SEDEF that mapped in the opposite orientation to within 500 kb of both inversion breakpoints. Next, we called repeats as SDs if they were duplicated within 500 kb of a breakpoint, were ≥1 kb in length,

had ≥70% identity with a duplication and had <70% of their sequence masked as common repeats. We then determined the density of SDs within 500 kb of each inversion breakpoint (note that we excluded breakpoints at chromosome ends as telomeres are not fully assembled in these genome assemblies). To compare the breakpoint SD density to that of random regions genome-wide, we also ran SEDEF on each contig from the *P. m. bairdii* PacBio genome assembly and called SDs. We then randomly selected 10,000 sites from across the genome and calculated the density of SDs within 500 kb of each site. Finally, we tested whether inversion breakpoints were significantly enriched for SDs relative to the 10,000 randomized genome-wide regions using the Kolmogorov–Smirnov test in R.

**Genes near inversion breakpoints.** We used the *P. m. bairdii* genome annotation (Pman2.1_chr_NCBI.corrected.merged-with-Apollo.Aug19.sorted_chr15.gff3) to explore whether inversion breakpoints disrupted annotated protein-coding genes. We tested whether the number of breakpoints disrupting gene sequence was expected by chance based on overall gene density using a binomial test; we calculated the gene density (including exons, introns and untranslated regions) to be 39% genome-wide and then used binom.test in R to perform a binomial test, with probability of success of 0.39.

**Inversion frequencies**
**Sampling populations across species range.** To characterize the frequencies and distributions of the inversions across the *P. maniculatus* range, we included 3–46 individuals from each of an additional eight populations, which, when combined with the initial populations, yielded a total of 218 mice from 13 populations. For five of the new eight populations (populations *a*, *b*, *f*, *i* and *k* in Fig. 4a; see Supplementary Table 1 for sample details), we extracted DNA from liver tissue and performed whole-genome resequencing (~10–15× coverage) and variant calling as described above. We also performed whole-genome resequencing for 11 *P. leucopus* samples and two *P. californicus* samples (see Supplementary Table 1 for sample details), which we also included in our variant calling pipeline. For three additional populations (populations *d*, *g* and *h* in Fig. 4a; see Supplementary Table 1 for sample details), we obtained publicly available exome-sequencing data[61] (NCBI: PRJNA528923) and mapped sequencing reads to the *P. maniculatus* reference genome with BWA-MEM. We then performed variant calling as described above, except that these samples were joint-genotyped separately from the whole-genome resequenced samples. We approximated the *P. maniculatus* range using the IUCN Red List of Threatened Species (https://www.iucnredlist.org/species/16672/22360898) and plotted the range map in R, as shown in Fig. 4a.

**Phylogenetic trees.** To reconstruct the evolutionary relationships among populations, we used RAxML[62] to build maximum-likelihood trees. First, we created a tree for the five focal *P. maniculatus* populations and two outgroups (*Peromyscus leucopus* and *P. californicus*). Using hard-filtered SNPs from across the autosomes, we thinned SNPs to at most one SNP per 100 kb using vcftools and merged vcfs across chromosomes. We converted the merged vcf to a PHYLIP matrix using vcf2phylip.py (https://github.com/edgardomortiz/vcf2phylip) and removed invariant sites using ascbias.py (https://github.com/btmartin721/raxml_ascbias), resulting in a total of 12,292 SNPs. We then ran RAxML v.8.2.12 using the ASC_GTRCAT model with the conditional likelihood method, -asc-corr=lewis, to correct for the ascertainment bias due to using SNPs[63]. We ran 100 bootstraps, with '-f a' to perform rapid bootstrap analysis and visualized trees in iTOL[64]. We next created a tree for all 13 *P. maniculatus* populations and the two outgroups. To do so, we first merged the variants called for the three exome-sequenced populations with the whole-genome resequenced vcfs and subset each population to at most 15 individuals. We removed variants with missing genotypes for >20% of samples and masked inversion regions using

bcftools. We then converted the vcf to a PHYLIP matrix and removed invariant sites as described above, resulting in a total of 15,518 SNPs. We ran RAxML as described above, with 100 bootstraps using '-f a' to perform rapid bootstrap analysis and visualized trees in iTOL.

**Genotyping samples for inversions.** To genotype individuals for the presence or absence of inversions, we used a PCA approach. For each inversion, we selected closely related populations segregating for the inversion of interest and performed PCA for that inversion region using scikit-allel, as described above. PCA was performed with only a subset of populations to allow for the inversion of interest (rather than population divergence) to drive variance along PC1. We then projected the remaining samples onto the PC1 and PC2 axes. We genotyped samples for each inversion based on loading scores along PC1 (along which samples clustered into inversion genotype groups) with manual determination of boundaries. We verified that samples called as inversion heterozygotes had elevated heterozygosity in the inversion region using the count_het function in scikit-allel. We set any populations with ambiguous clustering along PC1 for a given inversion to have missing genotypes. Finally, we determined inversion genotype frequencies for each population and tested for deviations from HWE using HWE.chisq in R from the genetics package.

We also determined inversion genotypes for: (1) 547 $F_2$ hybrids from the *P. m. rubidus* × *P. m. gambelii* cross; and (2) the 136 wild-caught mice from the environmental transect. To do so, we first created a set of SNPs fixed between the inversion and standard arrangements using homozygous samples from only forest (*P. m. rubidus*) and prairie (*P. m. gambelii*) populations, unless there were fewer than three homozygous samples per genotype, in which case we included additional homozygous samples from nearby populations (populations *b* and *f*, Fig. 4a) to improve filtering. Previously, the $F_2$ hybrids were sequenced using the ddRAD-sequencing pipeline (as described in Recombination rates, NCBI: PRJNA687993), and the 136 transect mice were whole-genome resequenced at low coverage (NCBI: PRJNA688305)[20]. Using these sequencing data, we selected the fixed inversion-standard SNPs from bam files for the $F_2$ hybrids and transect mice using mpileup and performed the hidden Markov model step of the multiplexed shotgun genotyping pipeline[65] to determine genotype for each inversion.

**Mutational load.** To test whether the inversions were enriched for deleterious mutations compared with standard haplotypes, we analyzed the number of segregating nonsynonymous ($pN$) versus synonymous ($pS$) sites and nucleotide diversity at nonsynonymous ($\pi_N$) versus synonymous ($\pi_S$) sites using PopGenome[66]. For each inversion, we selected samples homozygous for the inversion arrangement and used readVCF to import biallelic SNPs for the samples and inversion regions of interest into PopGenome; specifically, we selected homozygous samples from the major *P. maniculatus* clade (Fig. 4b; populations *a*, *b*, *c*, *e*, *f*, *i* and *j*) except for inv10.0 and inv11.0, for which we also included populations *k*, *l* and *m* in order to sample both homozygous genotypes. We then used the set.synnonsyn function with the *P. m. bairdii* genome annotation to determine nonsynonymous and synonymous sites. Next, we computed nucleotide diversity for each synonymous and nonsynonymous site with the diversity.stats function. Finally, for 500 kb windows across each inversion region, we calculated the ratios $pN/pS$ and $\pi_N/\pi_S$ (using only sites that were segregating within the homozygous sample set). We then repeated these analyses for samples homozygous for the standard arrangement. To test whether the inversion and standard haplotypes significantly differed in $pN/pS$ or $\pi_N/\pi_S$, we performed two-sided *t*-tests in R. Inv7.1 was excluded from this analysis because we had sequencing data for only one homozygous inversion sample; inv9.1 was also excluded because it harbours only six genes.

We also tested whether the inverted or standard haplotypes were enriched for deleterious mutations compared to the rest of the genome. To do so, we included all samples from the major *P. maniculatus* clade

(Fig. 4b; populations *a*, *b*, *c*, *e*, *f*, *i* and *j*) and calculated $pN/pS$ and $\pi_N/\pi_S$ for 500 kb windows across all regions genome-wide, excluding the inverted regions. We tested whether the inverted or standard haplotypes showed significantly higher $pN/pS$ or $\pi_N/\pi_S$ than genome-wide regions using one-sided *t*-tests in R.

**SLiM simulations.** To explore a possible role of selection on the inversions, we performed forward-genetic simulations in SLiM v.3.6 (ref. [35]). We simulated the forest (population *c*, *P. m. rubidus*) and prairie (population *e*, *P. m. gambelii*) populations evolving under a previously estimated best-fit demographic model[20] and introduced an inversion as a Mendelian locus as a single copy. We set separate selection coefficients for the inversion locus in the forest versus prairie populations, varying the selection coefficients from −0.01 to +0.01. We introduced the inversion into either the forest or the prairie population at five time points, corresponding to $1.5 \times 10^4$, $1.5 \times 10^5$, $7.5 \times 10^5$, $1.5 \times 10^6$ and $2.2 \times 10^6$ generations ago, with $2.2 \times 10^6$ being the estimated time of the forest–prairie split. To reduce computational time, we scaled parameters by a factor of 100, with population sizes ($N$) and times divided by 100 (for example, after scaling, time points ranged from $1.5 \times 10^2$ to $2.2 \times 10^4$ generations ago) and migration rates ($m$) and selection coefficients ($s$) multiplied by 100 (for example, after scaling, selection coefficients ranged from −1.0 to +1.0), to keep $Nm$ and $Ns$ consistent[35]. For each set of forest and prairie selection coefficients and each time point, we ran 1,000 simulations and recorded the frequency of the inversion in the forest and prairie populations at the end of the simulation. Finally, for each scenario, we computed the probability that the inversion reached an absolute allele frequency difference between the forest and prairie populations >50%. All selection coefficients are reported as their values before scaling.

**Clinal variation.** To test whether inversion frequency was associated with local habitat, we analyzed *P. maniculatus* mice previously collected across a forest–prairie environmental gradient, which included 136 samples from nine sites across the Cascade mountains in Oregon, USA[20]. Using publicly available sequencing data[20] (NCBI: PRJNA688305), we genotyped the 136 samples for the inversions (see Genotyping samples for inversions section above) and then used the package HZAR v.0.2.5 (ref. [67]) to fit clines to inversion genotypes (https://github.com/oharring/chr15_inversion). We fit ten different cline models by varying the scaling of minimum and maximum allele frequencies (scaling: 'fixed' or 'free') and how exponential tails were fit (tails: 'none', 'left', 'right', 'mirror' and 'both'). We selected the best model for each inversion using Akaike information criterion (with correction for small sample sizes) (AICc) values. Clines shown in Fig. 6b are fit with tails: 'none' and scales: 'fixed'; best-fit clines are shown in Extended Data Fig. 8.

**Genotype–phenotype associations.** Using data from a reciprocal intercross between *P. m. rubidus* (forest population) × *P. m. gambelii* (prairie population) $F_2$ hybrids ($n = 547$) as described above, we tested for associations between inversion genotype and three forest-ecotype-defining traits: tail length, foot length and coat colour. We used previously published phenotypic measurements[20] and the inversion genotypes reported here. For each of the 13 polymorphic forest–prairie inversions, we tested whether inversion genotype was significantly correlated with trait variation using linear models in R, with genotype coded numerically (additive genetic model); for tail and foot length, we included body length as a fixed effect. We corrected for multiple hypothesis testing (that is, testing 13 different inversions) using Bonferroni correction.

**Reporting summary**
Further information on research design is available in the Nature Research Reporting Summary linked to this article.

## Data availability

Sequencing data are available from NCBI SRA under BioProject accessions PRJNA856879, PRJNA816517, PRJNA860096, PRJNA862503; NCBI SRA accessions for individual samples are listed in Supplementary Table 1. Source data are provided with this paper.

## Code availability

The code used for the analyses is available from GitHub (https://github.com/oharring/pman_inversions).

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

## Acknowledgements

We thank T. Sackton, D. Khost and members of the Hoekstra laboratory for their advice on the analyses; T. Sackton, J. Mallet, L. Gozashti, A. Kautt and members of the Mallet laboratory for providing helpful feedback on the manuscript; T. B. Wooldridge for sharing short-read sequencing data; and E. Hager and T. B. Wooldridge for many helpful discussions on inversions. The Bauer Core Facility at Harvard University provided short-read library preparation and sequencing services. The University of Washington PacBio Sequencing Core provided long-read library preparation and sequencing services. Computational analyses were run on the Odyssey and Cannon clusters supported by the Faculty of Arts and Sciences Research Computing Group at Harvard University. We thank the Museum of Southwestern Biology (University of New Mexico), Museum of Comparative Zoology (Harvard University), S. Cushman (US Forest Service, Rocky Mountain Research Station) and C. Thompson (University of Michigan) for providing specimens used in this study. O.S.H. was supported by a National Science Foundation Graduate Research Fellowship, a Harvard Quantitative Biology Student Fellowship (DMS 1764269), the Molecular Biophysics Training Grant (NIH NIGMS T32GM008313), an American Society of Mammalogists Grants-in-Aid of Research and a Society for the Study of Evolution R.C. Lewontin Early Award. H.E.H. is funded as a Howard Hughes Medical Institute Investigator.

## Author contributions

O.S.H. conceived the study and performed the analyses, with input from H.E.H. O.S.H. and H.E.H. wrote the manuscript.

## Competing interests

The authors declare no competing interests.

## Additional information

**Extended data** is available for this paper at https://doi.org/10.1038/s41559-022-01890-0.

**Correspondence and requests for materials** should be addressed to Olivia S. Harringmeyer or Hopi E. Hoekstra.

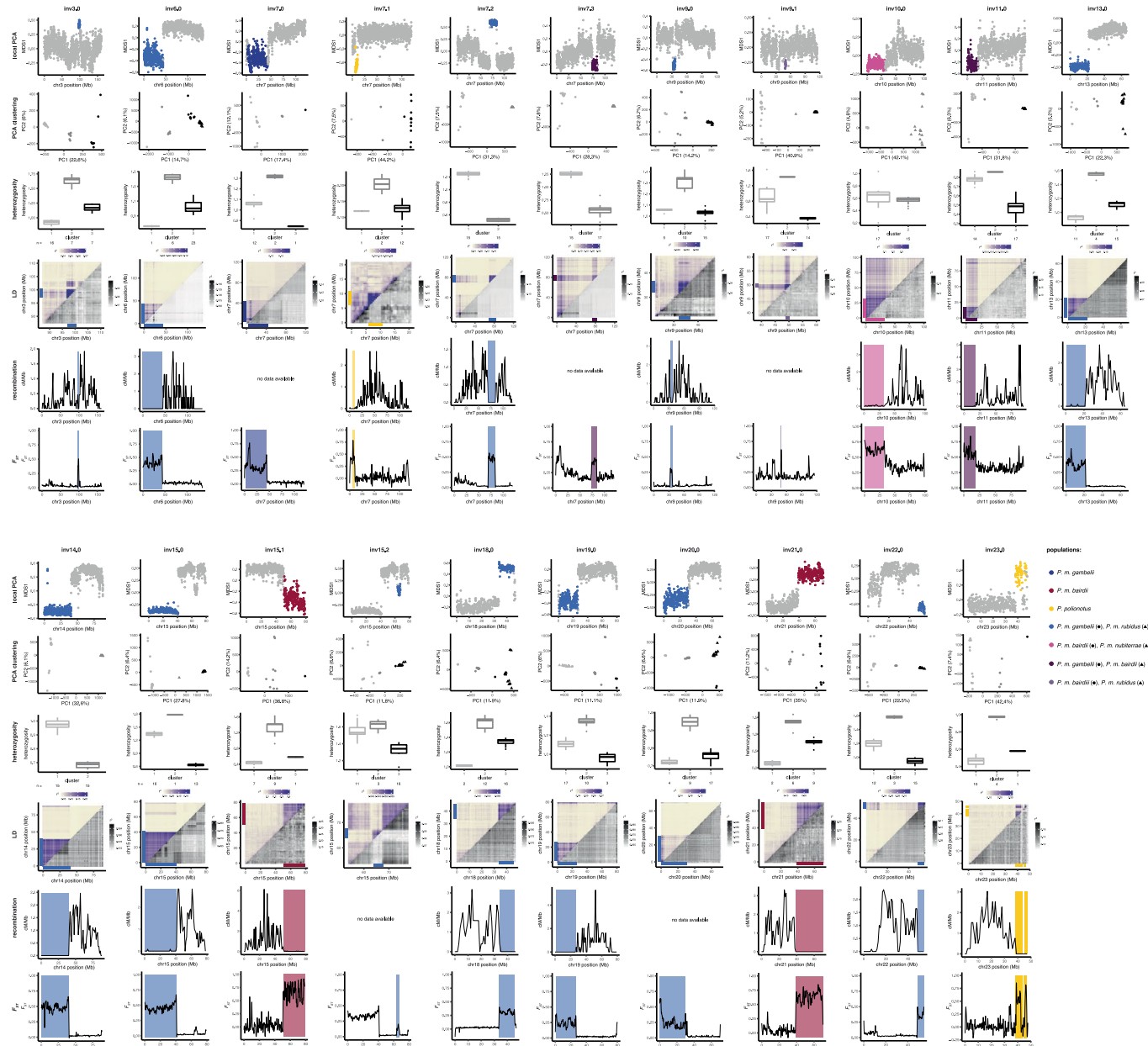

**Extended Data Fig. 1 | Identifying inversion polymorphisms based on population genomic signatures.** For each identified inversion polymorphism, the following signatures of inversions are shown (colors correspond to focal population or population-pair in which inversion was identified, see legend): (1) Local PCA performed with *lostruct*, where each dot represents a 100-kb window. Distances between local PCA maps are represented by the MDS1 axis, with outlier windows highlighted in color. (2) Clustering of samples by PCA for entire outlier region found with local PCA, assigned using k-means clustering. (3) Heterozygosity (percent of sites that are heterozygous) of outlier region for samples by cluster assignments from PCA above. Boxes indicate upper and lower quartiles; center line represents median; whiskers extend to minimum and maximum values within 1.5x interquartile range; points show outliers beyond

whiskers. Sample sizes are shown below the x-axis for each cluster. (4) LD for chromosomes harboring the example inversions, shown as mean $r^2$ values for paired windows across each chromosome. Upper triangle shows mean $r^2$ values including all samples from PCA clustering. Lower triangle shows mean $r^2$ values for only the more common homozygote genotype as determined in PCA clustering. Colored bars highlight outlier region from local PCA. Scales for $r^2$ values provided. (5) Recombination rates in cM/Mb shown for lab-born inversion heterozygotes. Outlier region found in local PCA is highlighted. Five inversions have missing data since inversion heterozygotes were not measured in the lab. (6) $F_{ST}$ between homozygous genotypes (clusters 1 and 3 from PCA and heterozygosity plots). Outlier regions found in local PCA are highlighted. Note that the discontinuity for inv23.0 is likely due to reference genome mis-assembly.

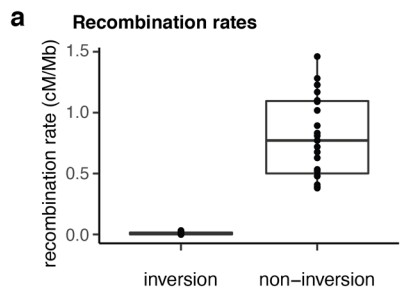

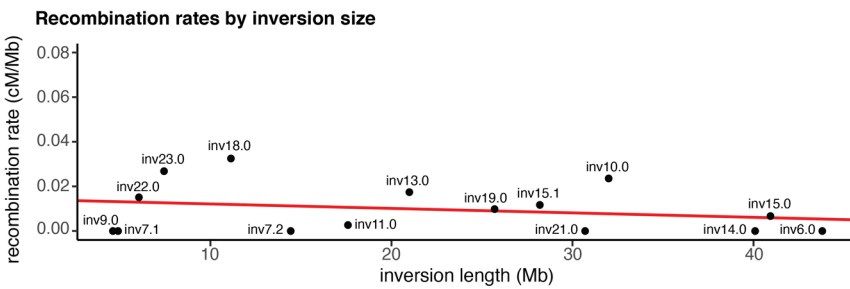

**Extended Data Fig. 2 | Recombination effects of inversion heterozygotes.**
(**a**) Recombination rates for inversion versus non-inversion regions from lab-born F$_2$ hybrids. Recombination rates for inversion regions are measured in inversion heterozygotes only; rates for non-inversion regions include all lab-born F$_2$ hybrids. Boxes indicate upper and lower quartiles; center line represents median; whiskers extend to minimum and maximum values within 1.5x interquartile range. Points for inversion regions represent inversions (n = 15); points for non-inversion regions represent chromosomes (excluding inversion regions) (n = 23). Recombination rate for inversion regions = 0.01 ± 0.03; non-inversion regions = 0.80 ± 0.34 (mean ± sd). (**b**) Recombination rates for inversion regions by inversion size in megabases. Linear fit between inversion length and recombination rate shown as red line (F-statistic on 1 and 13 degrees of freedom, p > 0.05).

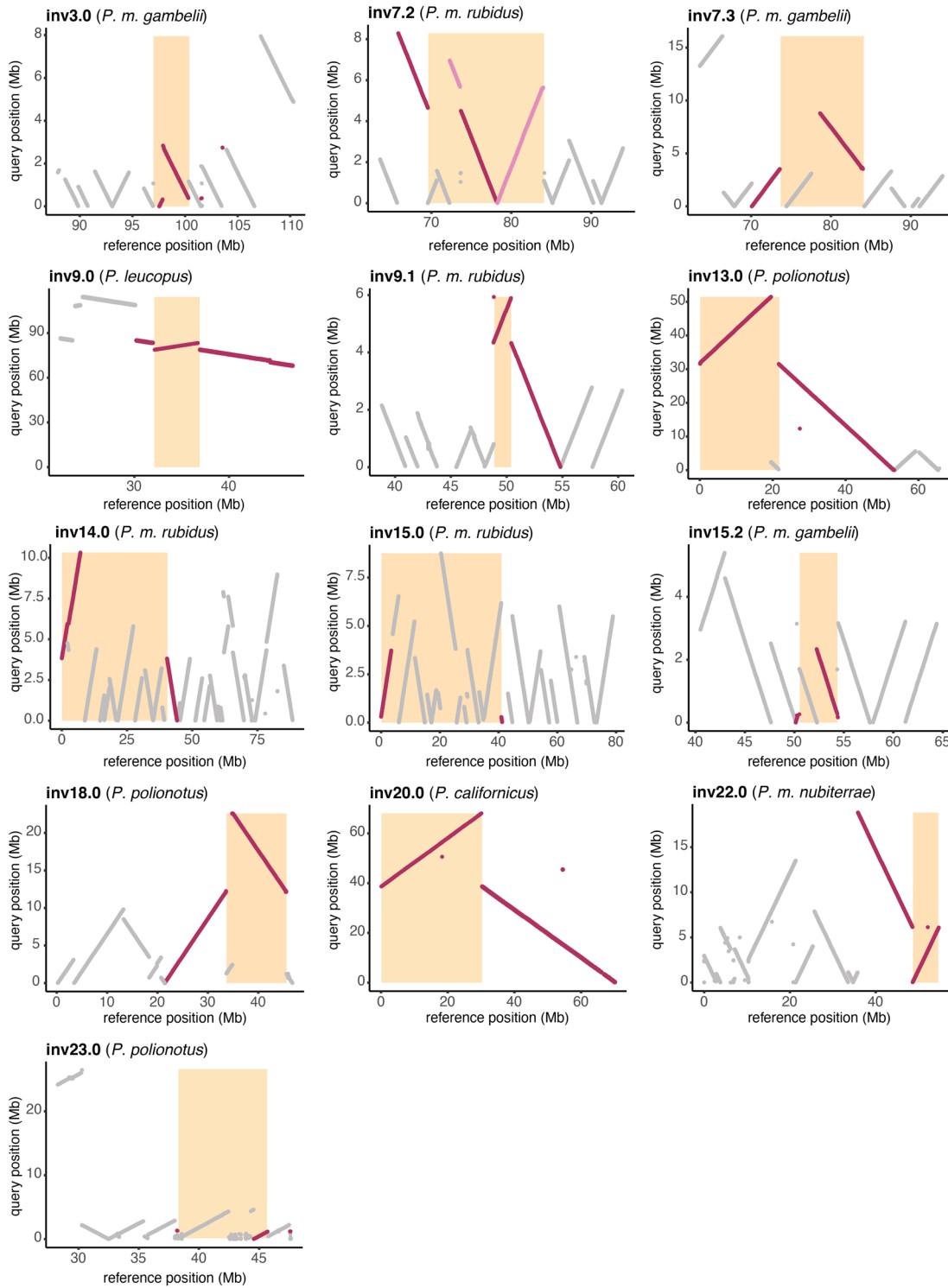

**Extended Data Fig. 3 | Localizing inversion breakpoints.** Contigs highlighting breakpoints for 13 inversions. Contigs from *de novo* genome assemblies ('query', y-axis) were aligned to the *P. maniculatus* reference genome ('reference', x-axis) with *nucmer*. Populations corresponding to the *de novo* assembly used in each plot are given; for inv9.0 and inv20.0 inversions, breakpoints were localized by aligning *P. leucopus* and *P. californicus* reference genomes to the *P. maniculatus* reference genome, respectively. Contigs (gray) and those identifying inversion breakpoints (red) are shown. Predicted inversion boundaries are highlighted (orange box). For the inv7.2 plot, the pink contig highlights a derived inversion (inv7.3) in the reference genome; when the reference genome is re-oriented to the ancestral state, the contig highlighted in red shows the inv7.2 inversion breakpoints.

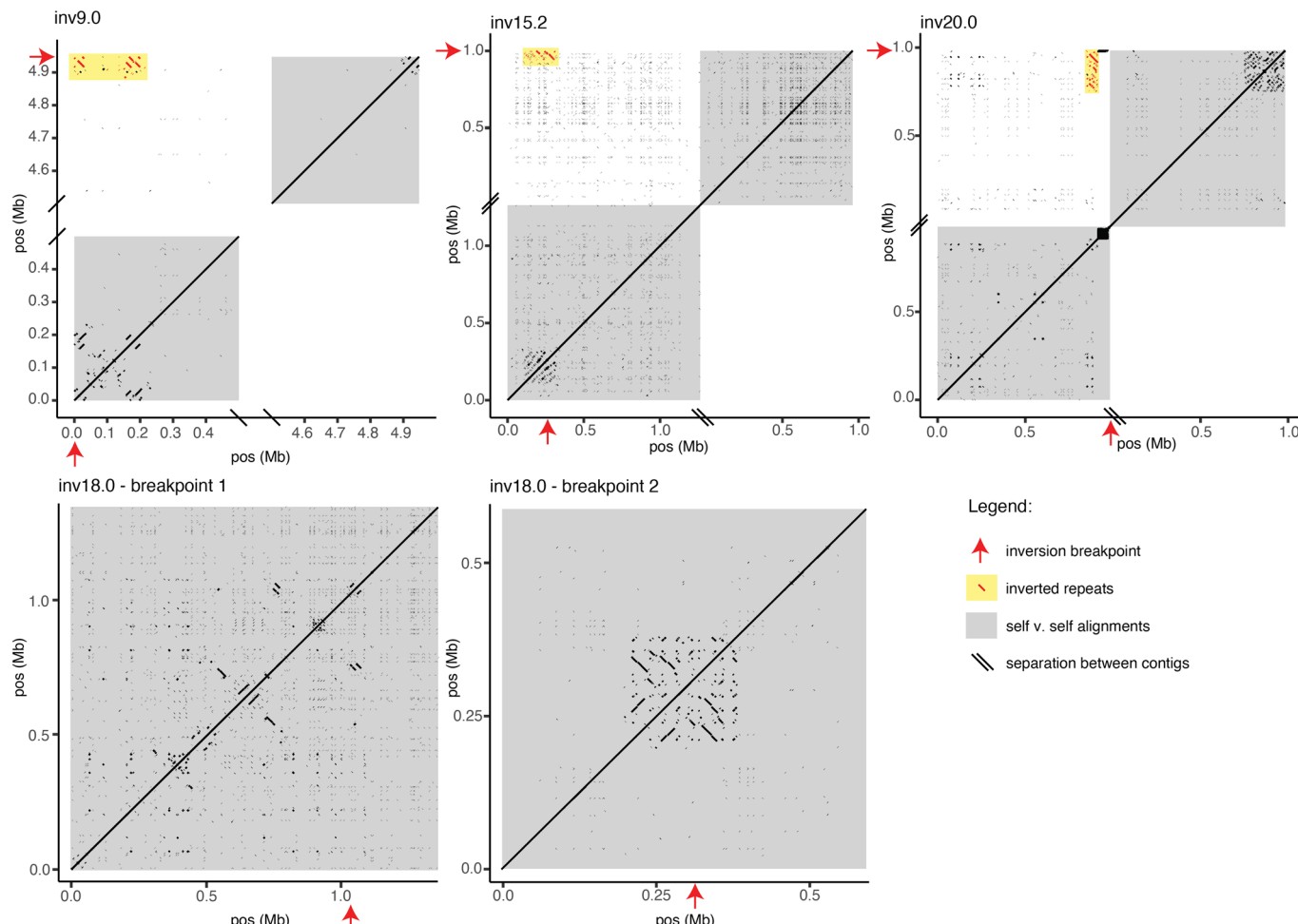

**Extended Data Fig. 4 | Inverted repeats and segmental duplications.**
Examples of inversion breakpoints near large inverted repeats (inv9.0, inv15.2, inv20.0) and segmental duplications (inv9.0, inv15.2, inv20.0, inv18.0). Dotplots show alignments for long-read assembly contigs spanning or nearly spanning breakpoints. Self-v-self alignments are highlighted (gray boxes), with alignments between breakpoint regions (within 500 kb of breakpoints) shown in upper left quadrant for inv9.0, inv15.2 and inv20.0. Location of breakpoints (red arrows) shown; only alignments with length >100 bp and within 500 kb of the breakpoints are shown. Inverted repeats mapping to within 500 kb of both breakpoints are shown (red) and highlighted (yellow boxes). Self-v-self alignments also show segmental duplications near breakpoints.

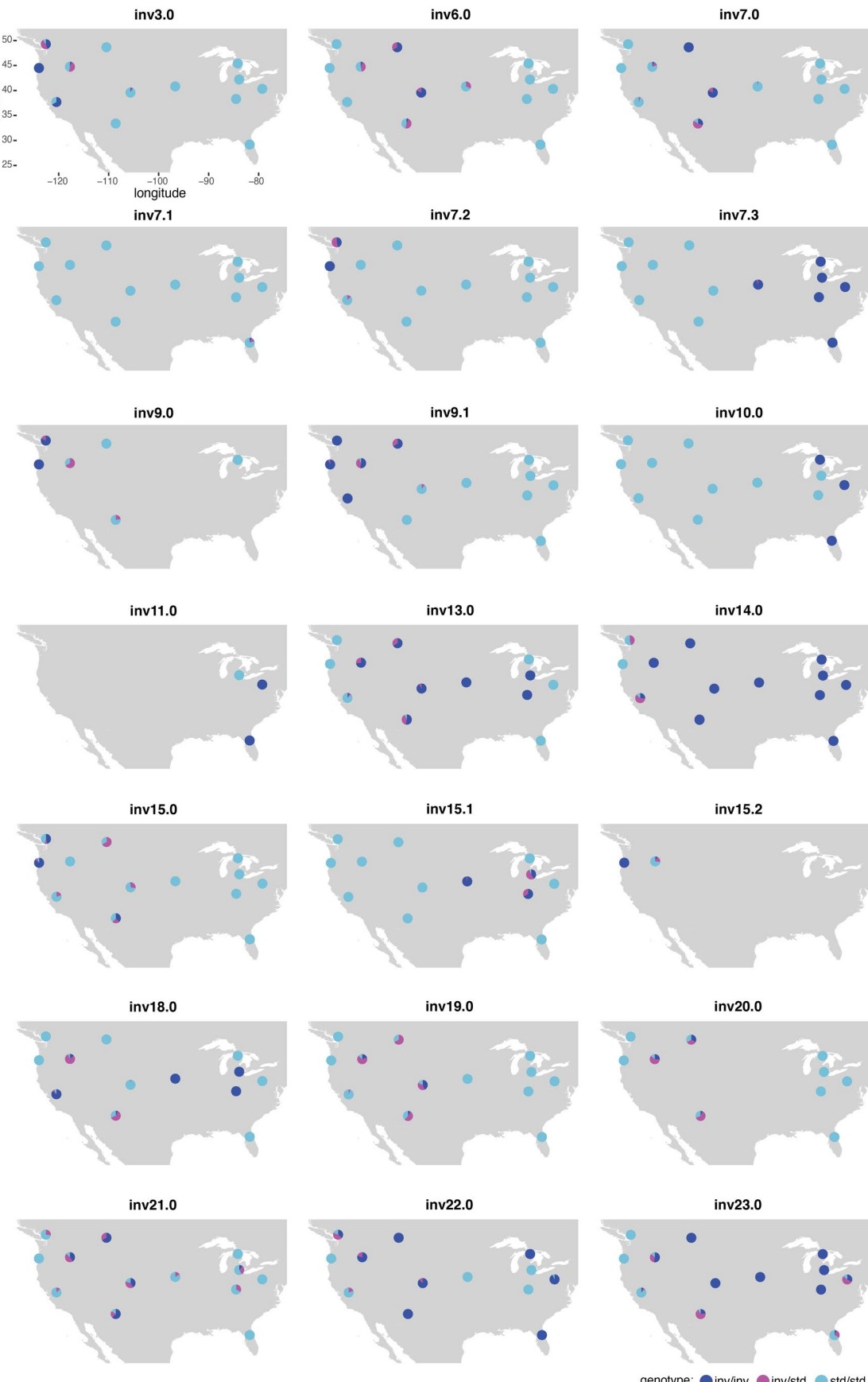

**Extended Data Fig. 5 | Distributions of inversions across species range.** Genotype frequencies shown across species range for each inversion (for 13 populations shown in Fig. 4a). Inversions were genotyped with PCA; populations with ambiguous genotypes for a given inversion are not included.

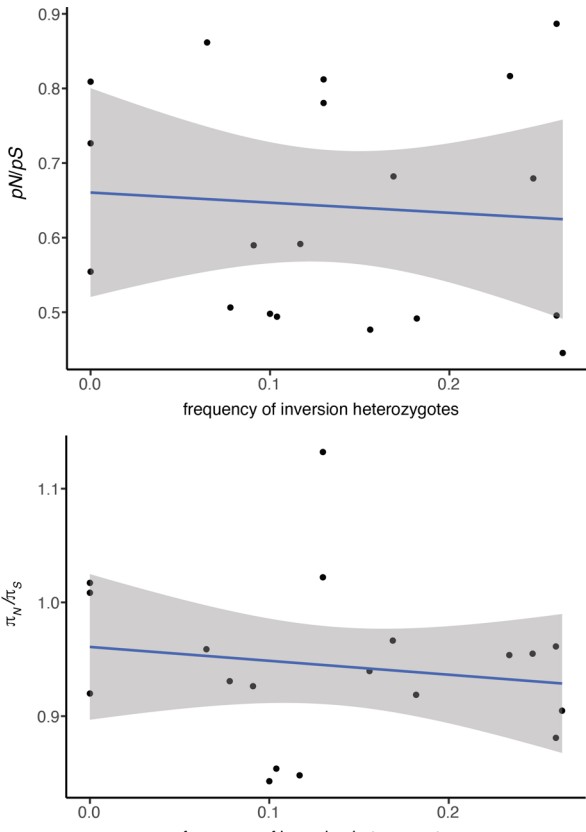

**Extended Data Fig. 6 | Mutational load by inversion heterozygote frequency.** The mean mutational load per inversion, as measured by *pN/pS* and $\pi_N/\pi_S$, is shown versus the frequency of inversion heterozygotes. Neither *pN/pS* nor $\pi_N/\pi_S$ are significantly correlated with the frequency of inversion heterozygotes (linear fits, shown as blue lines with 95% confidence intervals as gray shading; F-statistics on 1 and 17 degrees of freedom; p = 0.75 (top), p = 0.53 (bottom)).

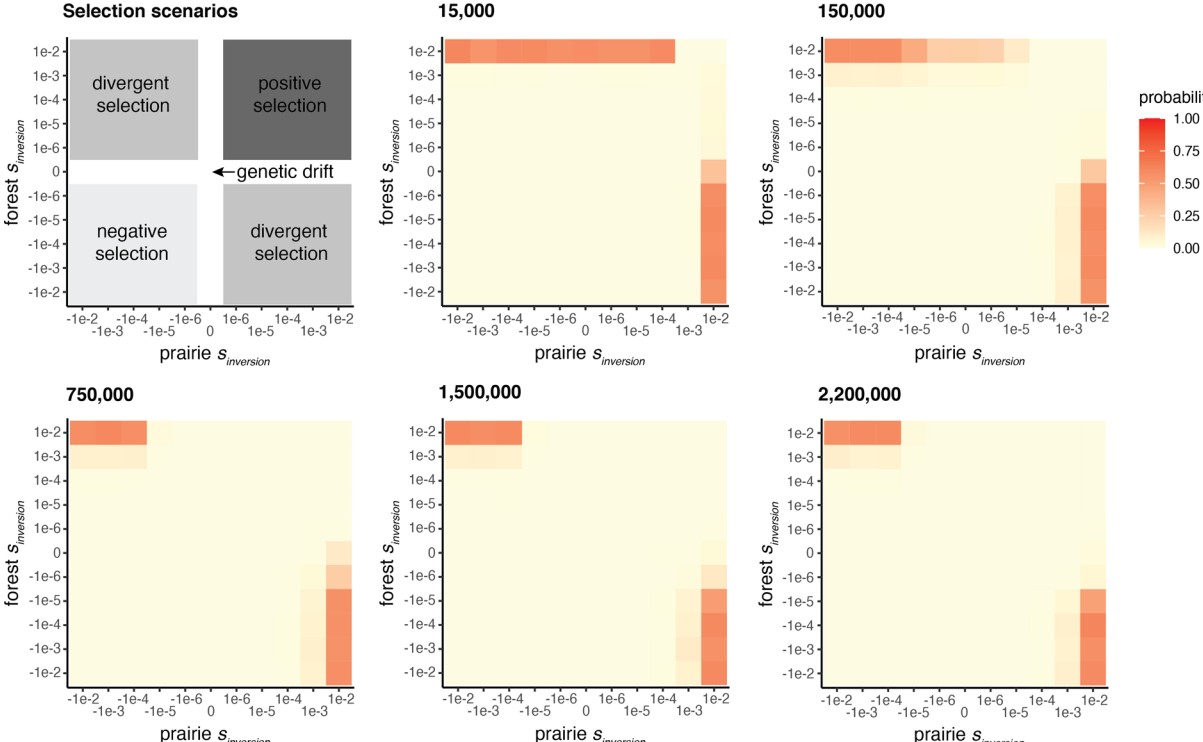

**Extended Data Fig. 7 | Forward genetic simulations of selection on inversions.** The evolution of an inversion was simulated in SLiM under a best-fitting demographic model of the forest and prairie populations. The inversion locus was introduced into the populations at five timepoints (shown in generations ago); the final timepoint (2.2 m generations) represents the forest-prairie split time estimate. Simulations for a range of selection coefficients for the inversion were performed, with 1,000 simulations per scenario. The selection scenarios are shown (upper left). Heatmaps show the probability of the inversion reaching a forest-prairie allele frequency difference >50% for each combination of selection coefficients.

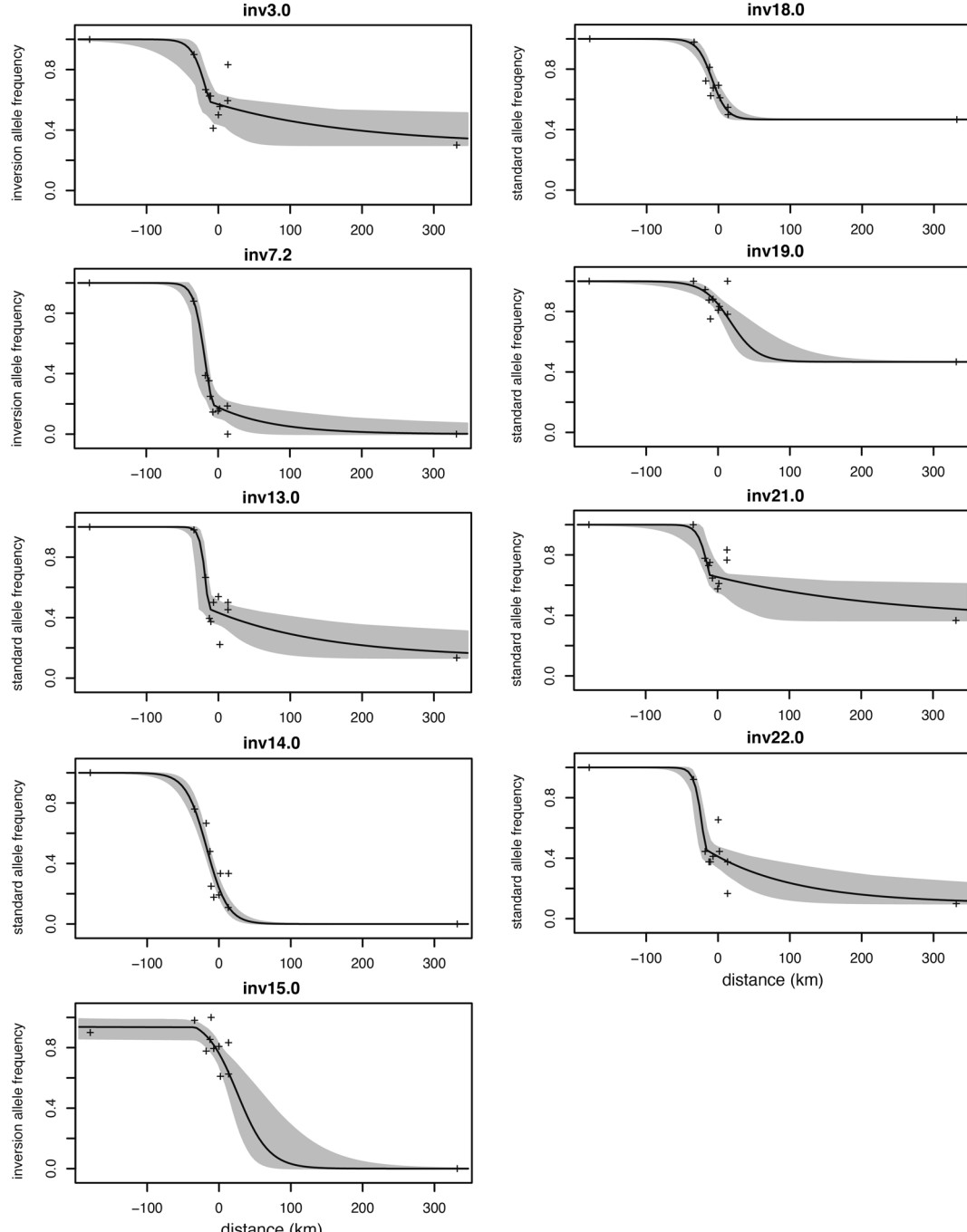

**Extended Data Fig. 8 | Clinal variation in inversion frequencies.** Inversion genotype frequencies shown across an environmental transect, with clines fit using *hzar*. Best-fit clines with 95% credible cline region shown. Sampled populations are highlighted (black crosses), with focal forest (left-most) and prairie (right-most) populations. Major allele in forest population is plotted, with y-axis label indicating *inversion* or *standard* haplotype.

**Extended Data Table 1 | Summary statistics for de novo genome assemblies with PacBio long-read sequencing**

| Population | PacBio subread N50 (kb) | Flye assembly total length (Gb) | Flye assembly contig N50 (Mb) | Flye assembly # of contigs |
|---|---|---|---|---|
| *P. maniculatus bairdii* | 36.9 | 2.53 | 11.2 | 5,747 |
| *P. maniculatus gambelii* | 36.6 | 2.60 | 3.2 | 12,590 |
| *P. maniculatus rubidus* | 37.9 | 2.61 | 2.6 | 15,237 |
| *P. maniculatus nubiterrae* | 40.2 | 2.56 | 5.5 | 8,512 |
| *P. p. subgriseus* | 38.8 | 2.51 | 9.5 | 3,108 |

Contig-level genomes were assembled using *flye* for one individual from each of the five focal populations. PacBio sequencing outputs are reported as subread N50 and *flye* genome assemblies are summarized with total length of assemblies, contig N50s, and number of contigs. Differences in assembly contiguity are likely driven by differences in heterozygosity in the sequenced samples.

**Extended Data Table 2 | Inversion genotypes for long-read samples**

| Inversion | *P. m. rubidus* | *P. m. gambelii* | *P. m. bairdii* | *P. m. nubiterrae* | *P. p. subgriseus* |
|---|---|---|---|---|---|
| inv3.0 | 1/1 | 0/0 | 0/0 | 0/0 | 0/0 |
| inv6.0 | 0/0 | 0/1 | 0/0 | 0/0 | 0/0 |
| inv7.0 | 0/0 | 0/0 | 0/0 | 0/0 | 0/0 |
| inv7.1 | 0/0 | 0/0 | 0/0 | 0/0 | 0/0 |
| inv7.2 | 1/1 | 0/0 | 0/0 | 0/0 | 0/0 |
| inv7.3 | 0/0 | 0/0 | 1/1 | 1/1 | 1/1 |
| inv9.0 | 1/1 | 0/0 | NA | NA | NA |
| inv9.1 | 0/1 | 1/1 | 0/0 | 0/0 | 0/0 |
| inv10.0 | 0/0 | 0/0 | 0/0 | 1/1 | 1/1 |
| inv11.0 | NA | NA | 0/0 | 1/1 | 1/1 |
| inv13.0 | 0/0 | 1/1 | 1/1 | 0/0 | 0/0 |
| inv14.0 | 0/0 | 1/1 | 1/1 | 1/1 | 1/1 |
| inv15.0 | 1/1 | 0/0 | 0/0 | 0/0 | 0/0 |
| inv15.1 | 0/0 | 0/0 | 1/1 | 0/0 | 0/0 |
| inv15.2 | 1/1 | 0/0 | NA | NA | NA |
| inv18.0 | 0/0 | 1/1 | 1/1 | 0/0 | 0/0 |
| inv19.0 | 0/0 | 0/1 | 0/0 | 0/0 | 0/0 |
| inv20.0 | 0/0 | 0/1 | 0/0 | 0/0 | 0/0 |
| inv21.0 | 0/0 | 0/1 | 0/1 | 0/0 | 0/0 |
| inv22.0 | 0/0 | 0/1 | 0/0 | 1/1 | 1/1 |
| inv23.0 | 0/0 | 0/1 | 1/1 | 1/1 | 0/0 |

Inversion genotypes (0=standard, 1=inversion) for all 21 inversions in each of the five PacBio long-read sequenced samples. Inv6.0, inv7.0, inv7.1, inv19.0, inv20.0, inv21.0 are not represented by both homozygous genotypes in these samples.

# Reporting Summary

## Statistics

For all statistical analyses, confirm that the following items are present in the figure legend, table legend, main text, or Methods section.

| n/a | Confirmed | |
|---|---|---|
| ☐ | ☒ | The exact sample size (*n*) for each experimental group/condition, given as a discrete number and unit of measurement |
| ☐ | ☒ | A statement on whether measurements were taken from distinct samples or whether the same sample was measured repeatedly |
| ☐ | ☒ | The statistical test(s) used AND whether they are one- or two-sided *Only common tests should be described solely by name; describe more complex techniques in the Methods section.* |
| ☐ | ☒ | A description of all covariates tested |
| ☐ | ☒ | A description of any assumptions or corrections, such as tests of normality and adjustment for multiple comparisons |
| ☐ | ☒ | A full description of the statistical parameters including central tendency (e.g. means) or other basic estimates (e.g. regression coefficient) AND variation (e.g. standard deviation) or associated estimates of uncertainty (e.g. confidence intervals) |
| ☐ | ☒ | For null hypothesis testing, the test statistic (e.g. *F*, *t*, *r*) with confidence intervals, effect sizes, degrees of freedom and *P* value noted *Give P values as exact values whenever suitable.* |
| ☒ | ☐ | For Bayesian analysis, information on the choice of priors and Markov chain Monte Carlo settings |
| ☒ | ☐ | For hierarchical and complex designs, identification of the appropriate level for tests and full reporting of outcomes |
| ☒ | ☐ | Estimates of effect sizes (e.g. Cohen's *d*, Pearson's *r*), indicating how they were calculated |

*Our web collection on statistics for biologists contains articles on many of the points above.*

## Software and code

Policy information about availability of computer code

| Data collection | Short-read whole-genome sequencing was performed using Illumina NovaSeq. Long-read whole-genome sequencing was performed using PacBio Sequel II. |
|---|---|
| Data analysis | BWA v0.7.17 was used for read-mapping, and GATK v3.8 was used for variant calling. Lostruct (https://github.com/petrelharp/local_pca) was used for local PCA. Population genomic analyses were performed using scikit-allel v1.3.2 (https://github.com/cggh/scikit-allel). bcftools v1.5 and vcftools v0.1.15 were used for variant filtering. Linkage disequilibrium was computed using the script emerald2windowldcounts.pl (https://github.com/owensgl/reformat, https://github.com/owensgl/haploblocks). Genome assembly was performed using flye v2.8.3 (https://github.com/fenderglass/Flye). Variant calling from long-read data was performed with ngmlr (https://github.com/philres/ngmlr) and longshot (https://github.com/pjedge/longshot). Mummer v3 was used for genome alignments. SEDEF (https://github.com/vpc-ccg/sedef) was used for detecting segmental duplications. RepeatMasker v4.0.5 was used to identify and mask common repeats. Phylogenetic trees were created with RAxML v8.2.12, with data prepared using vcf2phylip.py (https://github.com/edgardomortiz/vcf2phylip) and ascbias.py (https://github.com/btmartin721/raxml_ascbias). The genetics (https://cran.r-project.org/web/packages/genetics/index.html) package in R was used to analyze genotype frequencies. PopGenome v2.7.5 was used for mutational load analyses. SLiM v3.6 was used for forward-genetic simulations. HZAR v0.2.5 was used for clinal analyses. Additional data analyses and plotting were performed in R v3.4.2. |

For manuscripts utilizing custom algorithms or software that are central to the research but not yet described in published literature, software must be made available to editors and reviewers. We strongly encourage code deposition in a community repository (e.g. GitHub). See the Nature Portfolio guidelines for submitting code & software for further information.

## Data

Policy information about <u>availability of data</u>

All manuscripts must include a <u>data availability statement</u>. This statement should provide the following information, where applicable:

- Accession codes, unique identifiers, or web links for publicly available datasets
- A description of any restrictions on data availability
- For clinical datasets or third party data, please ensure that the statement adheres to our <u>policy</u>

Sequencing data is available from NCBI SRA (project numbers PRJNA856879, PRJNA816517, PRJNA860096, PRJNA862503).

# Field-specific reporting

Please select the one below that is the best fit for your research. If you are not sure, read the appropriate sections before making your selection.

☒ Life sciences    ☐ Behavioural & social sciences    ☐ Ecological, evolutionary & environmental sciences

For a reference copy of the document with all sections, see nature.com/documents/nr-reporting-summary-flat.pdf

# Life sciences study design

All studies must disclose on these points even when the disclosure is negative.

| | |
|---|---|
| Sample size | Sample sizes for short-read whole genome re-sequencing (n=15-17 samples/population, 15x coverage) enable genotype calls and estimates of population genomic parameters at reasonable costs. Sample sizes for long-read sequencing (n=1 sample/population) were chosen based on feasibility. |
| Data exclusions | No data were excluded from this study. |
| Replication | All analyses pipelines are fully described in the methods, and both associated data and code are provided. |
| Randomization | Randomization is not relevant for this study because experimental groups were not compared in this study. |
| Blinding | Blinding is not relevant for this study because samples were not assigned to experimental groups. |

# Reporting for specific materials, systems and methods

We require information from authors about some types of materials, experimental systems and methods used in many studies. Here, indicate whether each material, system or method listed is relevant to your study. If you are not sure if a list item applies to your research, read the appropriate section before selecting a response.

### Materials & experimental systems

| n/a | Involved in the study |
|---|---|
| ☒ ☐ | Antibodies |
| ☒ ☐ | Eukaryotic cell lines |
| ☒ ☐ | Palaeontology and archaeology |
| ☐ ☒ | Animals and other organisms |
| ☒ ☐ | Human research participants |
| ☒ ☐ | Clinical data |
| ☒ ☐ | Dual use research of concern |

### Methods

| n/a | Involved in the study |
|---|---|
| ☒ ☐ | ChIP-seq |
| ☒ ☐ | Flow cytometry |
| ☒ ☐ | MRI-based neuroimaging |

## Animals and other organisms

Policy information about <u>studies involving animals</u>; <u>ARRIVE guidelines</u> recommended for reporting animal research

| | |
|---|---|
| Laboratory animals | Five laboratory mice (4 Peromyscus maniculatus, 1 Peromyscus polionotus, approximately 60-100 days old) were used in this study for long-read sequencing. One laboratory mouse (Peromyscus californicus) was used in this study for short-read whole-genome sequencing. |
| Wild animals | Wild-caught specimens and tissues used in this study were museum accessioned or obtained from previous publications. |
| Field-collected samples | This study did not involve samples directly collected from the field. |

Ethics oversight          Harvard Unviersity's IACUC approved of the experiments of this study.

Note that full information on the approval of the study protocol must also be provided in the manuscript.

