## [Peer Review File · Nature Ecology & Evolution]

Peer Review Information

Journal: Nature Ecology & Evolution

Manuscript Title: Chromosomal inversion polymorphisms shape the genomic landscape of deer mice

Corresponding author name(s): Olivia S. Harringmeyer, Hopi E. Hoekstra

Editorial Notes:

Reviewer Comments & Decisions:

Decision Letter, initial version:
--

7th June 2022

Dear Ms Harringmeyer,

Your manuscript entitled "Massive inversion polymorphisms shape the genomic landscape of deer mice" has now been seen by 3 reviewers, whose comments are attached. The reviewers have raised a number of concerns which will need to be addressed before we can offer publication in Nature Ecology & Evolution. We will therefore need to see your responses to the criticisms raised and to some editorial concerns, along with a revised manuscript, before we can reach a final decision regarding publication.

We therefore invite you to revise your manuscript taking into account all reviewer and editor comments. Please highlight all changes in the manuscript text file.

* If you have not done so already please begin to revise your manuscript so that it conforms to our Article format instructions at <http://www.nature.com/natecolevol/info/final-submission>. Refer also to any guidelines provided in this letter.

[REDACTED]

Nature Ecology & Evolution is committed to improving transparency in authorship. As part of our efforts in this direction, we are now requesting that all authors identified as 'corresponding author' on published papers create and link their Open Researcher and Contributor Identifier (ORCID) with their account on the Manuscript Tracking System (MTS), prior to acceptance. ORCID helps the scientific community achieve unambiguous attribution of all scholarly contributions. You can create and link your ORCID from the home page of the MTS by clicking on 'Modify my Springer Nature account'. For more information please visit www.springernature.com/orcid.

[REDACTED]

Reviewers' comments:

Reviewer #1 (Remarks to the Author):

In this manuscript, Harringmeyer and Hoekstra investigate the role of segregating chromosomal inversions in shaping genetic diversity, and in adaptation, in deer mouse. First, they identify regions

2whose patterns of diversity (local population structure, heterozygosity...) are consistent with expectations for polymorphic inversions, and confirm that recombination is suppressed in these regions. Using long read data, they then confirm and map breakpoints for most of these inversions, and explore sequence patterns at breakpoint regions. Finally, they examine possible deleterious effects associated with the presence of segregating inversions, and the role of several of these inversions in differentiating, and likely promoting adaptation in, forest and prairie ecotypes of deer mouse.

The manuscript is very well written and was a pleasure to read. While similar analyses and insights have been individually presented in other experimental systems, this work stands out for its thoroughness and comprehensiveness, painting a complete picture of the origin, diversity, and likely adaptive function of large inversions segregating in deer mice populations.

I have just a few comments and suggestion:

- Page 3, line 2: what is the rationale for performing this analysis in individual populations or population pairs, instead of across all the populations at once? Are populations too divergent to be meaningfully analyzed all together?

- Page 3, line 4: I am not sure this is the most appropriate reference – I think it would make more sense to cite lostruct (Li and Ralph 2019), or eventually Todesco et al. 2020 and/or Huang et al. 2020, which perform a very similar set of analyses to identify and confirm large segregating inversions.

- Page 5, line 35: I understand that comparing levels of non-synonymous substitutions between "inverted" (i.e. derived) and "standard" (i.e. ancestral) orientation at inversion region would detect eventual increases in deleterious mutation in the derived allele, which would have been present at lower frequencies at least in the period immediately following its emergence. However, another important comparison would be with non-inversion regions of the genome, which would tell whether either or both orientations of the inversion experience increased accumulation of deleterious mutations due to the reduction in their effective population size (as in Huang et al. 2022).

- Page 15, line 30: I am probably missing something, but here it says that the split between forest and prairie (I am interpreting this as the split between forest (c) and prairie (e) ecotypes of *P. maniculatus*) happened 2.2M generations ago. Hager et al., 2021 places instead the diverge at 10-15KYA. What is the difference between the two estimates? (while I am completely ignorant of *P. maniculatus* biology, I am assuming they reproduce less than 2,000 times/year 9). On a separate note, having divergence estimates for the different inversions (as done in Hager et al. 2021 for inv 15.0) would be of interest.

- Figure 3A: the thinner yellow and orange lines in the left panel (NHEJ) make it look like those are single-strand overhangs – is that the case (are those meant to represent microhomology regions)?

- Figure 4A: I am guessing the tan outline on the map represents the range of *P. maniculatus*?

- Figure 4B: It would be helpful to highlight the focal populations in red also in this panel, and possibly

3also, if it not too much information to cram together, the forest and prairie population (c and e), maybe using a tan and green background colour. Also, the black outlines for population not in HWE are hard to see, a brighter colour might make it easier.

Reviewer #2 (Remarks to the Author):

This manuscript is clear, well-written, and well-organized. The topic is of broad interest among evolutionary biologists and geneticists. The approach (at the genome scale) is still quite rare, and the findings provide an important reference point for future studies. The results are striking and motivate questions about the prevalence and evolutionary significance of inversions. My comments about the manuscript are few and relatively minor.

1. It would be helpful in the introduction to provide more detail on the results of studies to date to put this work in context, particularly refs 12-15. Doing so would highlight the unique contributions of this work and provide the reader with a clearer frame of reference for the results.
2. This is also true in the discussion. For example, on Page 7 ~line 40, there is a summary of the number and size of the inversions noting that 420 million bps experience suppressed recombination. This is noted as remarkable. A reminder for the reader of the % of the genome that is involved along with context on findings from what few systems do have data is needed. Relatedly, this is a very minor issue of wordsmithing, but if data on the "prevalence and significance of inversion polymorphisms remains largely elusive," how do we really know if this system is "remarkable"? What are our expectations and why?
3. Page 5, line 5: this first paragraph details the distribution and frequency of inversions. The statement "to uncover the evolutionary histories of these inversions, we next.." sets a higher expectation, however. I was expecting to see data on estimates of ages or origins of inversions or analysis of selection, etc. That may be out of scope, but rewording of this this section is needed to make it less speculative.
4. Page 5, line 27; "the other 11 inversions with localized breakpoints do not disrupt genes (Figure 5A), although their breakpoints may still influence gene expression. Thus, the majority.....inversions are unlikely to confer strongly deleterious effects due to their breakpoints." I agree with this sentiment, but this seems like a bit of an overstatement. Disruption of gene expression can be deleterious and there is evidence from the literature that inversions affect expression of nearby genes. It would be more correct to say that the 11 inversions that do not interrupt genes are less likely to convey strongly deleterious effects than the two that do interrupt genes. Also—what is known about the two genes that are interrupted in terms of function?
5. Page 5, Mutational load: While adding this analysis to this paper may be out of scope, another approach to investigating mutational load in inversions uses TEs (as in ref 30). It would be useful to discuss why this specific approach was chosen.
6. Discussion, Page 8, line 10: This hypothesis is intriguing and seems testable using a simulation approach. It would also be good to explicitly state something like "As with sunflowers (30), we hypothesize....." to acknowledge that both ref 30 and this paper suggest a similar explanation for inversions that show little evidence of mutational load.
7. One other minor wordsmithing issue to consider—the paper argues that "inversions substantially

4shape the recombination landscape” but also that inversion homozygosity is high enough to avoid load resulting from reduced recombination. Both of these statements can be true and are not necessarily in opposition, but there is an opportunity to be very clear and careful about how inversions can suppress recombination and have potential evolutionary impacts (especially when there is variation among populations and divergent selection) while still avoiding suppression within populations because of high homozygosity. This also seems like an opportunity to use simulations to determine how much the expected impacts of suppressed recombination are mitigated by homozygosity.

Reviewer #3 (Remarks to the Author):

Review of ‘massive inversion polymorphisms shape the genomic landscape of deer mice’

This is a well written manuscript on the role of inversion polymorphisms in shaping the genome of wild deer mice. Using a combination of short read illumina data (15x) and long read pacbio data, the authors describe naturally occurring inversion polymorphisms >1Mb in this species (5 populations, 4 deer and 1 oldfield mouse, 15 individuals per population), identify the location of breakpoints, quantify the mutational load of inversions and their putative significance in environmental adaptation. After having read the paper, I think the real value of the submitted work is in the careful description of the inversions, especially their exact chromosomal location relative to centromeres and telomeres and what can be derived from it. I think this should be featured more upfront. More specific comments are below.

Comments

The authors filter the dataset so that only inversions larger than 1 MB can be detected, and then they state that the study provides evidence for ‘massive’ inversions that are segregating in this species. Given the filtering, this is a little akin to a one tailed test, and given the authors never considered ‘small’ inversions, I think this general headline (its stated like this in the title, which should be corrected) and throughout the manuscript, should be amended to reflect how the search was conducted.

The genomic resources and knowledge available for this species are tremendous and provide real strength to this study. The authors build their inversion discovery work on previous cytogenetic studies. They cite refs 22 and 23, saying that several inversions had been described using cytogenetics. I would like the authors to expand on this and state how many inversions were found previously, and how that overlaps with their findings. Indeed, it is stated that out of the 21 inversions detected in this study, 20 are new, meaning there was only 1 overlap. How many were previously detected but were not found in this study, and why do the authors think they were able to discover 20 additional ones? Has it got to do with their low frequency in the population? More about detection bias and general overlap between studies would be useful.

The work attempts to link these inversions to adaptation, but the hypothesis testing framework that is described at the end of the Introduction lack rigour and detail. From reading just the Introduction, it appears that the authors merely seek to describe the occurrence and frequency of inversions, however, when reading the rest of the study it becomes clear that much more is attempted and achieved (1: detection of inversions, description of location and size, centromeres and telomeres,

5gene content, breakpoint description, 2: impact on recombination, and mutational load, 3: frequency of inversions in the wild, signals of adaptation and relationship to traits). I urge the authors to carefully describe their rationale and testing framework at the end of the Introduction to provide a more rigorous framework of relating inversions to evolutionary processes.

Given the size of the inverted regions, their impact on the 3d structure of chromosomes (which impacts the gene exposure/accessibility) and the relative distance to/from heterochromatin regions of genes within the inversion, will be strongly impacted. All of this will likely have a pronounced influence on the gene expression. I think that the authors should discuss this in much more detail.

What is the relationship between the mutational load in inversions that are overdominant, i.e. where inversion heterozygotes are common? Can this be plotted (frequency of hets vs mutational load)?

It is stated that inversions have a significant impact on some phenotypic traits, such as tail length.

Can more exact measures please be provided? What is meant by significant? 2% or 20%?

Page 7, line 40, add a to state 'with a mean length'

Future work should start to explore the role of smaller <1MB inversions I think. If the authors agree with this, then maybe this can be added to the Discussion.

*****END*****

Author Rebuttal to Initial comments

Reviewers' comments:

Reviewer #1 (Remarks to the Author):

In this manuscript, Harringmeyer and Hoekstra investigate the role of segregating chromosomal inversions in shaping genetic diversity, and in adaptation, in deer mouse. First, they identify regions whose patterns of diversity (local population structure, heterozygosity...) are consistent with expectations for polymorphic inversions, and confirm that recombination is suppressed in these regions. Using long read data, they then confirm and map breakpoints for most of these inversions, and explore sequence patterns at breakpoint regions. Finally, they examine possible deleterious effects associated with the presence of segregating inversions, and the role of several of these inversions in differentiating, and likely promoting adaptation in, forest and prairie ecotypes of deer mouse.

The manuscript is very well written and was a pleasure to read. While similar analyses and insights have been individually presented in other experimental systems, this work stands out for its thoroughness and comprehensiveness, painting a complete picture of the origin, diversity, and likely adaptive function of

6large inversions segregating in deer mice populations.

We thank the reviewer for these comments.

I have just a few comments and suggestion:

- Page 3, line 2: what is the rationale for performing this analysis in individual populations or population pairs, instead of across all the populations at once? Are populations too divergent to be meaningfully analyzed all together?

Yes. When we include all populations, population structure is driven by population divergence, which masks the signatures of inversions. Therefore, we included only individual populations or population pairs for this analysis, such that the inversion signatures were detectable.

- Page 3, line 4: I am not sure this is the most appropriate reference – I think it would make more sense to cite lostruct (Li and Ralph 2019), or eventually Todesco et al. 2020 and/or Huang et al. 2020, which perform a very similar set of analyses to identify and confirm large segregating inversions.

We revised the citation for this section, and now cite Huang et al. 2020 and Todesco et al. 2020.

- Page 5, line 35: I understand that comparing levels of non-synonymous substitutions between “inverted” (i.e. derived) and “standard” (i.e. ancestral) orientation at inversion region would detect eventual increases in deleterious mutation in the derived allele, which would have been present at lower frequencies at least in the period immediately following its emergence. However, another important comparison would be with non-inversion regions of the genome, which would tell whether either or both orientations of the inversion experience increased accumulation of deleterious mutations due to the reduction in their effective population size (as in Huang et al. 2022).

We added analyses of the whole-genome as suggested. In particular, we calculated pN/pS and π_N/π_S in genome-wide regions (excluding the inversion regions) and compared both the inversion and standard haplotypes to the genome-wide regions. We found that neither the inversions nor the standard haplotypes showed an enrichment of non-synonymous mutations relative to the rest of the genome. The results of this analysis are shown in Figure 5B and included in the text: “In addition, neither the inversions nor the standard haplotypes showed an enrichment for non-synonymous

7mutations (pN/pS and π_N/π_S) relative to the rest of the genome (one-sided t-test: $p>0.05$ for all inversions and standard haplotypes) (Figure 5B).” (page 6, lines 2-5).

- Page 15, line 30: I am probably missing something, but here it says that the split between forest and prairie (I am interpreting this as the split between forest (c) and prairie (e) ecotypes of *P. maniculatus*) happened 2.2M generations ago. Hager et al., 2021 places instead the diverge at 10-15KYA. What is the difference between the two estimates? (while I am completely ignorant of *P. maniculatus* biology, I am assuming they reproduce less than 2,000 times/year ϑ). On a separate note, having divergence estimates for the different inversions (as done in Hager et al. 2021 for inv 15.0) would be of interest.

Since the original posting of the Hager et al. 2021 pre-print, we have updated the analysis of the forest (population *c*) and prairie (population *e*) ecotypes. We tested a range of demographic models and found that the population histories were best fit by a model with an older split time (2.2M generations ago) with high levels of ongoing gene flow; this model outperformed our previous estimate of a recent split time, which did not account for migration rates. This updated analysis is included in the final version of the Hager et al. manuscript, which will be published later this month.

We agree that age estimates for the inversions are of interest. Using a strict molecular clock model, we estimated the inversion ages to range from 50 - 275 thousand years old based on divergence levels from the standard haplotypes. However, we are concerned that this model does not account for the complex evolutionary histories of the inversions. While we performed in depth analysis of the age of inv15.0 in Hager et al. 2021, we note that this age estimate relies on a demographic model specific for the forest and prairie populations, which cannot be used here due to the widespread nature of most of the identified inversions. Creating a well-supported demographic model for *all* populations in which the inversions occur would be difficult, hindered by the number of populations and their connectedness via gene flow. Thus, while we are exploring alternative methods to date the inversions, we believe that these analyses are beyond the scope of this paper and are best suited for a follow-up study.

- Figure 3A: the thinner yellow and orange lines in the left panel (NHEJ) make it look like those are single-strand overhangs – is that the case (are those meant to represent microhomology regions)?

In Figure 3A, the thin yellow and orange lines were meant to represent single-strand overhangs. However, based on this comment, we realized that this may add more confusion to the figure. To improve the interpretability of the figure, we removed the single-strand overhangs.

- Figure 4A: I am guessing the tan outline on the map represents the range of *P. maniculatus*?

Yes. We updated the Figure 4A legend to describe the tan outline on the map.

- Figure 4B: It would be helpful to highlight the focal populations in red also in this panel, and possibly also, if it not too much information to cram together, the forest and prairie population (c and e), maybe using a tan and green background colour. Also, the black outlines for population not in HWE are hard to see, a brighter colour might make it easier.

We agree and updated Figure 4B based on these suggestions.

Reviewer #2 (Remarks to the Author):

This manuscript is clear, well-written, and well-organized. The topic is of broad interest among evolutionary biologists and geneticists. The approach (at the genome scale) is still quite rare, and the findings provide an important reference point for future studies. The results are striking and motivate questions about the prevalence and evolutionary significance of inversions. My comments about the manuscript are few and relatively minor.

We thank the reviewer for these comments.

1. It would be helpful in the introduction to provide more detail on the results of studies to date to put this work in context, particularly refs 12-15. Doing so would highlight the unique contributions of this work and provide the reader with a clearer frame of reference for the results.

We agree and elaborated on the results from previous studies in the introduction to provide further context for our study: “Using these approaches, recent studies have revealed the abundance of inversion polymorphisms in a few species: for example, sunflowers harbor dozens of large (1 – 100

9Mb) inversion polymorphisms¹⁶ and humans harbor, on average, hundreds of inversion polymorphisms that affect more DNA basepairs in total than single nucleotide polymorphisms^{12,13}.” (page 2, lines 28-32).

2. This is also true in the discussion. For example, on Page 7 ~line 40, there is a summary of the number and size of the inversions noting that 420 million bps experience suppressed recombination. This is noted as remarkable. A reminder for the reader of the % of the genome that is involved along with context on findings from what few systems do have data is needed. Relatedly, this is a very minor issue of wordsmithing, but if data on the “prevalence and significance of inversion polymorphisms remains largely elusive,” how do we really know if this system is “remarkable”? What are our expectations and why?

We agree and updated this section of the discussion to clarify and contextualize the results. In particular, we included the percent of the genome affected by the inversions and clarified that the number and sizes of the inversions “seem striking in the context of recombination” to emphasize the large effects of these inversions on recombination, but we removed stating that the abundance of the inversions is remarkable on its own. We also added context to these results: “While these results are consistent with large inversions causing suppression of recombination in other species (e.g., quails, maize, and cod), whether inversion polymorphisms affect similar proportions of the genome in other species remains largely unknown” (page 8, lines 13-16).

3. Page 5, line 5: this first paragraph details the distribution and frequency of inversions. The statement “to uncover the evolutionary histories of these inversions, we next..” sets a higher expectation, however. I was expecting to see data on estimates of ages or origins of inversions or analysis of selection, etc. That may be out of scope, but rewording of this this section is needed to make it less speculative.

We revised this sentence: “To explore the distributions of these inversions, we next characterized their frequencies across the species range” (page 5, lines 15-16).

4. Page 5, line 27; “the other 11 inversions with localized breakpoints do not disrupt genes (Figure 5A), although their breakpoints may still influence gene expression. Thus, the majority.....inversions are unlikely to confer strongly deleterious effects due to their breakpoints.” I agree with this sentiment, but this seems like a bit of an overstatement. Disruption of gene expression can be deleterious and there is evidence from the literature that inversions affect expression of nearby genes. It would be more correct to

say that the 11 inversions that do not interrupt genes are less likely to convey strongly deleterious effects than the two that do interrupt genes. Also—what is known about the two genes that are interrupted in terms of function?

We edited this line to clarify the wording as suggested: “While these two inversions may affect phenotypes through disrupting gene function, the other 11 inversions with localized breakpoints do not disrupt genes (Figure 5A) and are thus less likely to convey strongly deleterious effects, although their breakpoints may still influence gene expression” (page 5, lines 37-40). The interrupted genes include *Slc39a5*, which is a zinc transporter that helps control intracellular zinc levels, *Baz2a*, which plays a role in histone binding activity, and *1700129C05Rik*, with unknown function.

5. Page 5, Mutational load: While adding this analysis to this paper may be out of scope, another approach to investigating mutational load in inversions uses TEs (as in ref 30). It would be useful to discuss why this specific approach was chosen.

We are very interested in characterizing TE abundance within the inversions and considered including this analysis in the manuscript. However, because the long-read genome assemblies presented here are at the contig-level (thus, are likely missing TE-rich genomic regions at contig boundaries), we decided to create chromosome-level genome assemblies for the inversion and standard haplotypes. These chromosome-level assemblies are still in progress, and we believe that the TE analysis is therefore best suited for a follow-up study. However, we revised the discussion to mention TE accumulation as an additional form of mutational load: “In deer mouse inversions, however, we did not find evidence for the accumulation of mutational load based on nonsynonymous mutations (although these inversions may harbor an excess of other types of deleterious variants such as transposable elements, which future work will further resolve).” (page 8, lines 31-34).

6. Discussion, Page 8, line 10: This hypothesis is intriguing and seems testable using a simulation approach. It would also be good to explicitly state something like “As with sunflowers (30), we hypothesize.....” to acknowledge that both ref 30 and this paper suggest a similar explanation for inversions that show little evidence of mutational load.

We agree and revised the text to include this citation suggestion: “As in sunflowers⁴⁷, we hypothesize that these inversions, which act as large-scale modifiers of recombination when heterozygous, largely evaded deleterious costs associated with suppressed recombination by quickly spreading to high frequencies in deer mice, whose large population sizes could facilitate effective purifying selection in inversion homozygotes³²” (page 8, lines 39-42). We also included an additional citation of Berdan et al. 2021, which provides support for this hypothesis. Through simulations, Berdan et al. find that deleterious mutation accumulation negatively correlates with haplotype frequency and population size; this study is consistent with the hypothesis that a short period of low inversion frequencies and the large population sizes of deer mice helped facilitate reduced mutational load in the inversions.

7. One other minor wordsmithing issue to consider—the paper argues that “inversions substantially shape the recombination landscape” but also that inversion homozygosity is high enough to avoid load resulting from reduced recombination. Both of these statements can be true and are not necessarily in opposition, but there is an opportunity to be very clear and careful about how inversions can suppress recombination and have potential evolutionary impacts (especially when there is variation among populations and divergent selection) while still avoiding suppression within populations because of high homozygosity. This also seems like an opportunity to use simulations to determine how much the expected impacts of suppressed recombination are mitigated by homozygosity.

We agree that this is an important point, and we edited the text in two places to clarify this point. First, in discussing the inversions’ effects on the recombination landscape, we added a statement on the importance of inversion frequency on recombination: “We found that the detected inversions substantially shape the recombination landscape of deer mice: while suppression of recombination is limited to inversion heterozygotes (so the frequency of an inversion will determine the extent to which it affects recombination), most deer mouse inversions are widespread and inversion heterozygotes are common in natural populations.” (page 8, lines 19-23). Second, we brought these two concepts together in our discussion of mutational load: “As in sunflowers⁴⁷, we hypothesize that these inversions, which act as large-scale modifiers of recombination when heterozygous, largely evaded deleterious costs associated with suppressed recombination by quickly spreading to high frequencies in deer mice, whose large population sizes could facilitate effective purifying selection in inversion homozygotes³².” (page 8, lines 39-42).

Reviewer #3 (Remarks to the Author):

12Review of ‘massive inversion polymorphisms shape the genomic landscape of deer mice’

This is a well written manuscript on the role of inversion polymorphisms in shaping the genome of wild deer mice. Using a combination of short read illumina data (15x) and long read pacbio data, the authors describe naturally occurring inversion polymorphisms >1Mb in this species (5 populations, 4 deer and 1 oldfield mouse, 15 individuals per population), identify the location of breakpoints, quantify the mutational load of inversions and their putative significance in environmental adaptation.

After having read the paper, I think the real value of the submitted work is in the careful description of the inversions, especially their exact chromosomal location relative to centromeres and telomeres and what can be derived from it. I think this should be featured more upfront.

More specific comments are below.

Comments

The authors filter the dataset so that only inversions larger than 1 MB can be detected, and then they state that the study provides evidence for ‘massive’ inversions that are segregating in this species. Given the filtering, this is a little akin to a one tailed test, and given the authors never considered ‘small’ inversions, I think this general headline (its stated like this in the title, which should be corrected) and throughout the manuscript, should be amended to reflect how the search was conducted.

We agree and revised the text based on this suggestion. In particular, we changed the title to “Chromosomal inversion polymorphisms shape the genomic landscape of deer mice”. In addition, we revised the discussion of inversion size, to emphasize the limitations of our methods: “Furthermore, since our approach was limited to detecting inversions >1 Mb in length, there are possibly many additional inversions of shorter lengths segregating within deer mice, which is an important direction for future work.” (page 8, lines 16-18).

The genomic resources and knowledge available for this species are tremendous and provide real strength to this study. The authors build their inversion discovery work on previous cytogenetic studies. They cite refs 22 and 23, saying that several inversions had been described using cytogenetics. I would like the authors to expand on this and state how many inversions were found previously, and how that overlaps with their findings. Indeed, it is stated that out of the 21 inversions detected in this study, 20 are new, meaning there was only 1 overlap. How many were previously detected but were not found in this study, and why do the authors think they were able to discover 20 additional ones? Has it got to do with their low frequency in the population? More about detection bias and general overlap between studies would be useful.

Previous cytogenetic studies found evidence for at least 13 large chromosomal rearrangements segregating within deer mice. We now note this in the introduction: “Early cytogenetic work in deer mice identified at least 13 visible chromosomal rearrangements^{23,24}.” (page 2, lines 36-37). However, the nature of these chromosomal rearrangements remained unknown. Our findings of 21 chromosomal inversions likely overlaps with some of these 13 previously identified rearrangements; however, due to changes in chromosome numbering, we cannot determine exactly which inversions correspond to which previously identified rearrangements. We explain this in the text: “Returning to this system in the molecular age, we detect 21 large inversion polymorphisms segregating within deer mice (some of which likely overlap with rearrangements detected via cytogenetics).” (page 2, lines 37-39). In addition, we originally stated that 20 of the 21 inversions are novel; in this statement, we intended to imply novelty of these inversions in the molecular age (since inv15.0 was recently described in our 2021 bioRxiv preprint). We appreciate the reviewer pointing out this statement, and the potential confusion, so we removed mention of novelty from this paragraph.

The work attempts to link these inversions to adaptation, but the hypothesis testing framework that is described at the end of the Introduction lack rigour and detail. From reading just the Introduction, it appears that the authors merely seek to describe the occurrence and frequency of inversions, however, when reading the rest of the study it becomes clear that much more is attempted and achieved (1: detection of inversions, description of location and size, centromeres and telomeres, gene content, breakpoint description, 2: impact on recombination, and mutational load, 3: frequency of inversions in the wild, signals of adaptation and relationship to traits). I urge the authors to carefully describe their rationale and testing framework at the end of the Introduction to provide a more rigorous framework of relating inversions to evolutionary processes.

We agree and elaborated our introduction to improve the set-up for the rest of the study. At the end of the introduction, we now describe in more detail our approach: “In localizing these inversions, we determine their positions relative to centromeres and telomeres, explore their effects on chromosome structure, characterize genomic content at their breakpoints, and propose a mechanism by which inversions arise in this species. Further, we quantify the impact of the inversions on recombination and the resulting effects on mutational load. Finally, we survey the inversions’ distributions across the species range and identify several inversions that contribute to local adaptation. Together, this work reveals proximate and ultimate mechanisms involved in the establishment and maintenance of inversion polymorphisms and suggests a prominent role for these inversions in local adaptation.” (page 2, lines 39-45; page 3, lines 1-2).

Given the size of the inverted regions, their impact on the 3d structure of chromosomes (which impacts

the gene exposure/accessibility) and the relative distance to/from heterochromatin regions of genes within the inversion, will be strongly impacted. All of this will likely have a pronounced influence on the gene expression. I think that the authors should discuss this in much more detail.

We added discussion of the possible effects on gene expression due to changes in chromosomal structure: “Furthermore, inversions also likely influence chromosome accessibility due to changes in three-dimensional genome structure, which, in addition to the mutations the inversions carry, may influence the expression of genes found within the inversions.” (pages 7, lines 32-34).

What is the relationship between the mutational load in inversions that are overdominant, i.e. where inversion heterozygotes are common? Can this be plotted (frequency of hets vs mutational load)?

We performed additional analyses to address this question. Specifically, we computed mean mutational load per inversion and tested for a correlation with inversion heterozygote frequencies. We did not find a significant correlation between mutational load (as measured by non-synonymous mutation accumulation) and the frequencies of inversion heterozygotes. These analyses are now included as a supplemental figure (Figure S6), and the results are noted in the text: “In addition, neither the inversions nor the standard haplotypes showed an enrichment for non-synonymous mutations (pN/pS and π_N/π_S) relative to the rest of the genome (one-sided t-test: $p > 0.05$ for all inversions and standard haplotypes) (Figure 5B), and we did not find a correlation between inversion heterozygote frequencies and mutational load (Figure S6).” (page 6, lines 2-6).

It is stated that inversions have a significant impact on some phenotypic traits, such as tail length. Can more exact measures please be provided? What is meant by significant? 2% or 20%?

To determine whether an inversion had a significant effect on a given phenotype, we used linear models (with Bonferroni correction to account for multiple hypothesis testing of different inversions; for details, see Methods). We clarified this section of the results to include more detail on the effect sizes (e.g., percent variance explained, additive effects) and significance testing: “Furthermore, five inversions (inv7.2, inv14.0, inv15.0, inv18.0, inv21.0) are significantly associated with an ecotype-defining trait, tail length, in lab-raised F_2 hybrids²⁰ ($p < 0.05$, linear model), and for all five, the forest arrangement is associated with longer tails (Figure 6C), consistent with long tails being important for balance in arboreal habitats³⁶. These five inversions together explain 22.7% of the variance in tail length (individually explaining 1.8 – 12.3% of the variance, with additive effects

15ranging from 1.1 – 2.7mm change in tail length). Inv15.0 was also previously found to be significantly associated with coat color, a second ecotype-defining trait²⁰ (explaining 40% of coat color variance) (Figure 6C).” (page 6, lines 43-46; page 7, lines 1-5).

Page 7, line 40, add a to state ‘with a mean length’

We updated the text as suggested.

Future work should start to explore the role of smaller <1MB inversions I think. If the authors agree with this, then maybe this can be added to the Discussion.

We agree and added a phrase describing the importance of exploring smaller inversions in future work: “Furthermore, since our approach was limited to detecting inversions >1 Mb in length, there are possibly many additional inversions of shorter lengths segregating within deer mice, which is an important direction for future work.” (page 8, lines 16-18).

Decision Letter, first revision:

14th July 2022

Dear Dr. Harringmeyer,

Thank you for submitting your revised manuscript "Chromosomal inversion polymorphisms shape the genomic landscape of deer mice" (NATECOLEVOL-220516448A). It has now been seen again by the original reviewers and their comments are below. The reviewers find that the paper has improved in revision, and therefore we'll be happy in principle to publish it in Nature Ecology & Evolution, pending minor revisions to satisfy the reviewers' final requests and to comply with our editorial and formatting guidelines.

16If the current version of your manuscript is in a PDF format, please email us a copy of the file in an editable format (Microsoft Word or LaTeX)-- we can not proceed with PDFs at this stage.

[REDACTED]

Reviewer #1 (Remarks to the Author):

The authors addressed all my (minor) comments satisfactorily, and I am happy to recommend publication of the manuscript.

I would just maybe recommend to the authors to include their answer to my first comment (asking why local PCAs were not done on the whole dataset but in population pairs) in the methods section for clarity.

Reviewer #2 (Remarks to the Author):

I feel the authors have addressed my concerns in this revision. They have re-worded sections to clarify and added additional analysis that supports their major conclusions. There are some opportunities to expand further on this work, but the authors have mentioned these in the text and I agree that they are beyond the scope of this paper. As is, the paper is a meaningful contribution to this research area and will be of great interest.

Reviewer #3 (Remarks to the Author):

I have gone over the revision and feel that the authors have done a good job and addressed my concerns on the previous version, as well as the comments made by the other reviewers. My only last comment that I would like to make is that I think the below rationale (which is a reply of the authors to one of the comments from Reviewer 1) should be included in the manuscript.

'When we include all populations, population structure is driven by population divergence, which masks the signatures of inversions. Therefore, we included only individual populations or population pairs for this analysis, such that the inversion signatures were detectable.'

I would like to congratulate the authors for their fine work.

Our ref: NATECOLEVOL-220516448A

26th July 2022

Dear Dr. Harringmeyer,

Thank you for your patience as we've prepared the guidelines for final submission of your Nature Ecology & Evolution manuscript, "Chromosomal inversion polymorphisms shape the genomic landscape of deer mice" (NATECOLEVOL-220516448A). Please carefully follow the step-by-step instructions provided in the attached file, and add a response in each row of the table to indicate the changes that you have made. Please also check and comment on any additional marked-up edits we have proposed within the text. Ensuring that each point is addressed will help to ensure that your revised manuscript can be swiftly handed over to our production team.

****We would like to start working on your revised paper, with all of the requested files and forms, as soon as possible (preferably within two weeks). Please get in contact with us immediately if you anticipate it taking more than two weeks to submit these revised files.****

In recognition of the time and expertise our reviewers provide to Nature Ecology & Evolution's editorial process, we would like to formally acknowledge their contribution to the external peer review of your manuscript entitled "Chromosomal inversion polymorphisms shape the genomic landscape of deer mice". For those reviewers who give their assent, we will be publishing their names alongside the published article.

Nature Ecology & Evolution offers a Transparent Peer Review option for new original research manuscripts submitted after December 1st, 2019. As part of this initiative, we encourage our authors to support increased transparency into the peer review process by agreeing to have the reviewer comments, author rebuttal letters, and editorial decision letters published as a Supplementary item. When you submit your final files please clearly state in your cover letter whether or not you would like to participate in this initiative. Please note that failure to state your preference will result in delays in

18accepting your manuscript for publication.

Cover suggestions

As you prepare your final files we encourage you to consider whether you have any images or illustrations that may be appropriate for use on the cover of Nature Ecology & Evolution.

Nature Ecology & Evolution has now transitioned to a unified Rights Collection system which will allow our Author Services team to quickly and easily collect the rights and permissions required to publish your work. Approximately 10 days after your paper is formally accepted, you will receive an email in providing you with a link to complete the grant of rights. If your paper is eligible for Open Access, our Author Services team will also be in touch regarding any additional information that may be required to arrange payment for your article.

Please note that *Nature Ecology & Evolution* is a Transformative Journal (TJ). Authors may publish their research with us through the traditional subscription access route or make their paper immediately open access through payment of an article-processing charge (APC). Authors will not be required to make a final decision about access to their article until it has been accepted. [Find out more about Transformative Journals](https://www.springernature.com/gp/open-research/transformative-journals)

Authors may need to take specific actions to achieve [compliance with funder and institutional open access mandates](https://www.springernature.com/gp/open-research/funding/policy-compliance-faqs). If your research is supported by a funder that requires immediate open access (e.g. according to [Plan S principles](https://www.springernature.com/gp/open-research/plan-s-compliance)) then you should select the gold OA route, and we will direct you to the compliant route where possible. For authors selecting the subscription publication route, the journal's standard licensing terms will need to be accepted, including [self-archiving-and-license-to-publish](https://www.nature.com/nature-portfolio/editorial-policies/self-archiving-and-license-to-publish). Those licensing terms will supersede any other terms that the author or any third party may assert apply to any version of the manuscript.

19Please note that you will not receive your proofs until the publishing agreement has been received through our system.

For information regarding our different publishing models please see our [Transformational Journals](https://www.springernature.com/gp/open-research/transformational-journals) page. If you have any questions about costs, Open Access requirements, or our legal forms, please contact ASJournals@springernature.com.

[REDACTED]

[REDACTED]

Reviewer #1:

Remarks to the Author:

The authors addressed all my (minor) comments satisfactorily, and I am happy to recommend publication of the manuscript.

I would just maybe recommend to the authors to include their answer to my first comment (asking why local PCAs were not done on the whole dataset but in population pairs) in the methods section for clarity.

Reviewer #2:

Remarks to the Author:

I feel the authors have addressed my concerns in this revision. They have re-worded sections to clarify and added additional analysis that supports their major conclusions. There are some opportunities to expand further on this work, but the authors have mentioned these in the text and I agree that they are beyond the scope of this paper. As is, the paper is a meaningful contribution to this research area and will be of great interest.

Reviewer #3:

Remarks to the Author:

I have gone over the revision and feel that the authors have done a good job and addressed my concerns on the previous version, as well as the comments made by the other reviewers. My only last comment that I would like to make is that I think the below rationale (which is a reply of the authors

20to one of the comments from Reviewer 1) should be included in the manuscript.

'When we include all populations, population structure is driven by population divergence, which masks the signatures of inversions. Therefore, we included only individual populations or population pairs for this analysis, such that the inversion signatures were detectable.'

I would like to congratulate the authors for their fine work.

Author Rebuttal, first revision:Reviewer #1:

Remarks to the Author:

The authors addressed all my (minor) comments satisfactorily, and I am happy to recommend publication of the manuscript.

I would just maybe recommend to the authors to include their answer to my first comment (asking why local PCAs were not done on the whole dataset but in population pairs) in the methods section for clarity.

We thank the reviewer for the helpful comments on our manuscript. We added to the methods section a description of why local PCAs were performed in population pairs: “Note that when we include all populations, population structure is driven by population divergence, which masks the signatures of possible inversions. Therefore, we included only individual populations or population pairs for this analysis, such that inversion signatures were detectable” (pg. 10, lines 33-36).

Reviewer #2:

Remarks to the Author:

I feel the authors have addressed my concerns in this revision. They have re-worded sections to clarify and added additional analysis that supports their major conclusions. There are some opportunities to expand further on this work, but the authors have mentioned these in the text and I agree that they are beyond the scope of this paper. As is, the paper is a meaningful contribution to this research area and will be of great interest.

We thank the reviewer for the helpful comments on our manuscript.

Reviewer #3:

Remarks to the Author:

I have gone over the revision and feel that the authors have done a good job and addressed my concerns on the previous version, as well as the comments made by the other reviewers. My only last comment that I would like to make is that I think the below rationale (which is a reply of the authors to one of the comments from Reviewer 1) should be included in the manuscript.

'When we include all populations, population structure is driven by population divergence, which masks the signatures of inversions. Therefore, we included only individual populations or population pairs for this analysis, such that the inversion signatures were detectable.'

I would like to congratulate the authors for their fine work.

We thank the reviewer for the helpful comments on our manuscript. As noted above, we added this rationale to the methods section (pg. 10, lines 33-36).

Final Decision Letter:

17th August 2022

Dear Olivia,

I am writing in the temporary absence of my colleague, Patrick Goymer.

We are pleased to inform you that your Article entitled "Chromosomal inversion polymorphisms shape the genomic landscape of deer mice", has now been accepted for publication in Nature Ecology & Evolution.

Over the next few weeks, your paper will be copyedited to ensure that it conforms to Nature Ecology and Evolution style. Once your paper is typeset, you will receive an email with a link to choose the appropriate publishing options for your paper and our Author Services team will be in touch regarding any additional information that may be required

You will not receive your proofs until the publishing agreement has been received through our system

Due to the importance of these deadlines, we ask you please us know now whether you will be difficult to contact over the next month. If this is the case, we ask you provide us with the contact information (email, phone and fax) of someone who will be able to check the proofs on your behalf, and who will be available to address any last-minute problems . Once your paper has been scheduled for online publication, the Nature press office will be in touch to confirm the details.

Acceptance of your manuscript is conditional on all authors' agreement with our publication policies (see www.nature.com/authors/policies/index.html). In particular your manuscript must not be published elsewhere and there must be no announcement of the work to any media outlet until the publication date (the day on which it is uploaded onto our web site).

Please note that *Nature Ecology & Evolution* is a Transformative Journal (TJ). Authors may publish their research with us through the traditional subscription access route or make their paper immediately open access through payment of an article-processing charge (APC). Authors will not be required to make a final decision about access to their article until it has been accepted. [Find out more about Transformative Journals](https://www.springernature.com/gp/open-research/transformative-journals)

Authors may need to take specific actions to achieve  > **compliance with funder and institutional open access mandates.** If your research is supported by a funder that requires immediate open access (e.g. according to [Plan S principles](https://www.springernature.com/gp/open-research/plan-s-compliance)) then you should select the gold OA route, and we will direct you to the compliant route where possible. For authors selecting the subscription publication route, the journal's standard licensing terms will need to be accepted, including [a href="https://www.nature.com/nature-portfolio/editorial-policies/self-archiving-and-license-to-publish"](https://www.nature.com/nature-portfolio/editorial-policies/self-archiving-and-license-to-publish). Those licensing terms will supersede any other terms that the author or any third party may assert apply to any version of the manuscript.

An online order form for reprints of your paper is available at <https://www.nature.com/reprints/author-reprints.html> > <https://www.nature.com/reprints/author-reprints.html>. All co-authors, authors' institutions and authors' funding agencies can order reprints using the form appropriate to their geographical region.

We welcome the submission of potential cover material (including a short caption of around 40 words) related to your manuscript; suggestions should be sent to Nature Ecology & Evolution as electronic files (the image should be 300 dpi at 210 x 297 mm in either TIFF or JPEG format). Please note that such pictures should be selected more for their aesthetic appeal than for their scientific content, and that colour images work better than black and white or grayscale images. Please do not try to design a cover with the Nature Ecology & Evolution logo etc., and please do not submit composites of images related to your work. I am sure you will understand that we cannot make any promise as to whether any of your suggestions might be selected for the cover of the journal.

You can generate the link yourself when you receive your article DOI by entering it here: <http://authors.springernature.com/share> > <http://authors.springernature.com/share>.

24[REDACTED]

P.S. Click on the following link if you would like to recommend Nature Ecology & Evolution to your librarian <http://www.nature.com/subscriptions/recommend.html#forms>

** Visit the Springer Nature Editorial and Publishing website at http://editorial-jobs.springernature.com?utm_source=ejp_NEcoE_email&utm_medium=ejp_NEcoE_email&utm_campaign=ejp_NEcoE for more information about our career opportunities. If you have any questions please click [here](mailto:editorial.publishing.jobs@springernature.com).**